# RMIX: RISK-SENSITIVE MULTI-AGENT REINFORCEMENT LEARNING

## ABSTRACT

Centralized training with decentralized execution (CTDE) has become an important paradigm in multi-agent reinforcement learning (MARL). Current CTDE-based methods rely on restrictive decompositions of the centralized value function across agents, which decomposes the global Q-value into individual Q values to guide individuals' behaviours. However, such expected, i.e., risk-neutral, Q value decomposition is not sufficient even with CTDE due to the randomness of rewards and the uncertainty in environments, which causes the failure of these methods to train coordinating agents in complex environments. To address these issues, we propose RMIX, a novel cooperative MARL method with the Conditional Value at Risk (CVaR) measure over the learned distributions of individuals' Q values. Our main contributions are in three folds: (i) We first learn the return distributions of individuals to analytically calculate CVaR for decentralized execution; (ii) We then propose a dynamic risk level predictor for CVaR calculation to handle the temporal nature of the stochastic outcomes during executions; (iii) We finally propose risk-sensitive Bellman equation along with *Individual-Global-MAX* (IGM) for MARL training. Empirically, we show that our method significantly outperforms state-of-the-art methods on many challenging StarCraft II tasks, demonstrating significantly enhanced coordination and high sample efficiency. Demonstrative videos and results are available in this anonymous link: https://sites.google.com/view/rmix.

## 1 INTRODUCTION

Reinforcement learning (RL) has made remarkable advances in many domains, including arcade video games (Mnih et al., 2015), complex continuous robot control (Lillicrap et al., 2016) and the game of Go (Silver et al., 2017). Recently, many researchers put their efforts to extend the RL methods into multi-agent systems (MASs), such as urban systems (Singh et al., 2020), coordination of robot swarms (Hüttenrauch et al., 2017) and real-time strategy (RTS) video games (Vinyals et al., 2019). Centralized training with decentralized execution (CTDE) (Oliehoek et al., 2008; Kraemer & Banerjee, 2016) has drawn enormous attention via training policies of each agent with access to global trajectories in a centralized way and executing actions given only the local observations of each agent in a decentralized way. Empowered by CTDE, several MARL methods, including value-based and policy gradient-based, are proposed (Foerster et al., 2017a; Sunehag et al., 2017; Rashid et al., 2018; Son et al., 2019). These MARL methods propose decomposition techniques to factorize the global Q value either by structural constraints or by estimating state-values or inter-agent weights to conduct the global Q value estimation. Among these methods, VDN (Sunehag et al., 2017) and QMIX (Rashid et al., 2018) are representative methods that use additivity and monotonicity structure constraints, respectively. With relaxed structural constraints, QTRAN (Son et al., 2019) guarantees a more general factorization than VDN and QMIX. Some other methods include incorporating an estimation of advantage values (Wang et al., 2020a) and proposing a multi-head attention method to represent the global values (Yang et al., 2020).

Despite the merits, most of these works focus on decomposing the global Q value into individual Q values with different constraints and network architectures, but ignore the fact that the expected, i.e., risk-neutral, value decomposition is not sufficient even with CTDE due to the randomness of rewards and the uncertainty in environments, which causes the failure of these methods to train coordinating agents in complex environments. Specifically, these methods only learn the expected values over

returns (Rashid et al., 2018) and do not handle the high variance caused by events with extremely high/low rewards to agents but small probabilities, which cause the inaccurate/insufficient estimations of the future returns. Therefore, instead of expected values, learning distributions of future returns, i.e., Q values, is more useful for agents to make decisions. Recently, QR-MIX (Hu et al., 2020) decomposes the estimated joint return distribution (Bellemare et al., 2017; Dabney et al., 2018a) into individual Q values. However, the policies in QR-MIX are still based expected individual Q values. Even further, given that the environment is nonstationary from the perspective of each agent, each agent needs a more dynamic way to choose actions based on the return distributions, rather than simply taking the expected values. However, current MARL methods do not extensively investigate these aspects.

Motivated by the previous reasons, we intend to extend the risk-sensitive[1] RL (Chow & Ghavamzadeh, 2014; Keramati et al., 2020; Zhang et al., 2020) to MARL settings, where risk-sensitive RL optimizes policies with a risk measure, such as variance, power formula measure value at risk (VaR) and conditional value at risk (CVaR). Among these risk measures, CVaR has been gaining popularity due to both theoretical and computational advantages (Rockafellar & Uryasev, 2002; Ruszczyński, 2010). However, there are two main obstacles: i) most of the previous works focus on risk-neutral or static risk level in single-agent settings, ignoring the randomness of reward and the temporal structure of agents' trajectories (Dabney et al., 2018a; Tang et al., 2019; Ma et al., 2020; Keramati et al., 2020); ii) many methods use risk measures over Q values for policy execution without getting the risk measure values used in policy optimization in temporal difference (TD) learning, which causes the global value factorization on expected individual values to sub-optimal behaviours in MARL. We provide a detailed review of related works in Appendix A due to the limited space.

In this paper, we propose RMIX, a novel cooperative MARL method with CVaR over the learned distributions of individuals' Q values. Specifically, our contributions are in three folds: (i) We first learn the return distributions of individuals by using Dirac Delta functions in order to analytically calculate CVaR for decentralized execution. The resulting CVaR values at each time step are used as policies for each agent via $\arg \max$ operation; (ii) We then propose a dynamic risk level predictor for CVaR calculation to handle the temporal nature of stochastic outcomes during executions. The dynamic risk level predictor measures the discrepancy between the embedding of current individual return distributions and the embedding of historical return distributions. The dynamic risk levels are agent-specific and observation-wise; (iii) We finally propose risk-sensitive Bellman equation along with IGM for centralized training. The risk sensitive Bellman equation enables CVaR value update in a recursive form and can be trained with TD learning via a neural network. These also allow our method to achieve temporally extended exploration and enhanced temporal coordination, which are key to solving complex multi-agent tasks. Empirically, we show that RMIX significantly outperforms state-of-the-art methods on many challenging StarCraft II[TM2] tasks, demonstrating enhanced coordination in many *symmetric* & *asymmetric* and *homogeneous* & *heterogeneous* scenarios and revealing high sample efficiency. To the best of our knowledge, our work is the *first* attempt to investigate cooperative MARL with risk-sensitive policies under the Dec-POMDP framework.

## 2 PRELIMINARIES

In this section, we provide the notation and the basic notions we will use in the following. We consider the probability space $(\Omega, \mathcal{F}, \mathrm{Pr})$, where $\Omega$ is the set of outcomes (sample space), $\mathcal{F}$ is a $\sigma$-algebra over $\Omega$ representing the set of events, and $\mathrm{Pr}$ is the set of probability distributions. Given a set $\mathcal{X}$, we denote with $\mathscr{P}(\mathcal{X})$ the set of all probability measures over $\mathcal{X}$.

**DEC-POMDP** A fully *MARL* problem can be described as a *decentralised partially observable Markov decision process* (Dec-POMDP) (Oliehoek et al., 2016) which can be formulated as a tuple $\mathcal{M} = \langle \mathcal{S}, \mathcal{U}, \mathcal{P}, R, \Upsilon, O, \mathcal{N}, \gamma \rangle$, where $s \in \mathcal{S}$ denotes the true state of the environment. Each agent $i \in \mathcal{N} := \{1, ..., N\}$ chooses an action $u_i \in \mathcal{U}$ at each time step, giving rise to a joint action vector, $\boldsymbol{u} := [u_i]_{i=1}^N \in \mathcal{U}^N$. $\mathcal{P}(\boldsymbol{s}'|\boldsymbol{s}, \boldsymbol{u}) : \mathcal{S} \times \mathcal{U}^N \times \mathcal{S} \mapsto \mathcal{P}(\mathcal{S})$ is a Markovian transition function and governs all state transition dynamics. Every agent shares the same joint reward function $R(\boldsymbol{s}, \boldsymbol{u}) : \mathcal{S} \times \mathcal{U}^N \mapsto \mathcal{R}$, and $\gamma \in [0, 1)$ is the discount factor. Due to *partial observability*, each agent has individual partial observation $\upsilon \in \Upsilon$, according to some observation function $O(\boldsymbol{s}, i) : \mathcal{S} \times \mathcal{N} \mapsto \Upsilon$.

---

[1]"Risk" refers to the uncertainty of future outcomes (Dabney et al., 2018a).

[2]StarCraft II is a trademark of Blizzard Entertainment, Inc.

Each agent also has an action-observation history $\tau_i \in \mathcal{T} := (\Upsilon \times \mathcal{U})^*$, on which it conditions its stochastic policy $\pi_i(u_i|\tau_i) : \mathcal{T} \times \mathcal{U} \mapsto [0, 1]$.

**CVaR** CVaR is a coherent risk measure and enjoys computational properties (Rockafellar & Uryasev, 2002) that are derived for loss distributions in discreet decision-making in finance. It gains popularity in various engineering and finance applications. CVaR (as illustrated in Figure 1) is the expectation of values that are less equal than the $\alpha$-percentile value of the distribution over returns.

Formally, let $X \in \mathcal{X}$ be a bounded random variable with cumulative distribution function $F(x) = \mathscr{P}[X \leq x]$ and the inverse CDF is $F^{-1}(u) = \inf\{x : F(x) \geq u\}$. The *conditional value at risk (CVaR)* at level $\alpha \in (0, 1]$ of a random variable $X$ is then defined as $\text{CVaR}_\alpha(X) := \sup_\nu \{\nu - \frac{1}{\alpha}\mathbb{E}[(\nu - X)^+]\}$ (Rockafellar et al., 2000) when $X$ is a discrete random variable. Correspondingly, $\text{CVaR}_\alpha(X) = \mathbb{E}_{X \sim F}[X|X \leq F^{-1}(\alpha)]$ (Acerbi & Tasche, 2002) when $X$ has a continuous distribution. The $\alpha$-percentile value is value at risk (VaR). For ease of notation, we write CVaR as a function of the CDF $F$, $\text{CVaR}_\alpha(F)$.

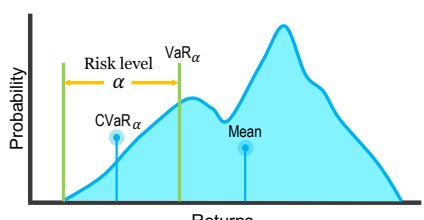

Figure 1: CVaR

**Risk-sensitive RL** Risk-sensitive RL uses risk criteria over policy/value, which is a sub-field of the Safety RL (García et al., 2015). Von Neumann & Morgenstern (1947) proposed the expected utility theory where a decision policy behaves as though it is maximizing the expected value of some utility functions. The theory is satisfied when the decision policy is a consistent and has a particular set of four axioms. This is the most pervasive notion of risk-sensitivity. A policy maximizing a linear utility function is called *risk-neutral*, whereas concave or convex utility functions give rise to *risk-averse* or *risk-seeking* policies, respectively. Many measures are used in RL such as CVaR (Chow et al., 2015; Dabney et al., 2018a) and power formula (Dabney et al., 2018a). However, few works have been done in MARL and it cannot be easily extended. Our work fills this gap.

**CTDE** CTDE has recently attracted attention from deep MARL to deal with nonstationarity while learning decentralized policies. One of the promising ways to exploit the CTDE paradigm is value function decomposition (Sunehag et al., 2017; Rashid et al., 2018; Son et al., 2019) which learns a decentralized utility function for each agent and uses a mixing network to combine these local Q values into a global action-value. It follows the IGM principle where the optimal joint actions across agents are equivalent to the collection of individual optimal actions of each agent (Son et al., 2019). To achieve learning scalability, existing CTDE methods typically learn a shared local value or policy network for agents.

## 3 METHODOLOGY

In this section, we present our framework RMIX, as displayed in Figure 2, where the agent network learns the return distribution of each agent, a risk operator network determines the risk level of each agent and the mixing network mixes the outputs of risk operators of agents to produce the global value. In the rest of this section, we first introduce the CVaR operator to analytically calculate the CVaR value with the modeled individual distribution of each agent in Section 3.1 and propose the dynamic risk level predictor to alleviate time-consistency issue in Section 3.2. Then, we introduce the risk-sensitive Bellman equation for both centralized training and decentralized execution in Section 3.3. Finally, we provide the details of centralized training of RMIX in Section 3.4. All proofs are provided in Appendix B.

### 3.1 CVAR OF RETURN DISTRIBUTION

In this section, we describe how we estimate the CVaR value. The value of CVaR can be either estimated through sampling or computed from the paremetrized return distribution (Rockafellar & Uryasev, 2002). However, the sampling method is usually computationally expensive (Tang et al., 2019). Therefore, we let each agent learn a return distribution parameterized by a mixture

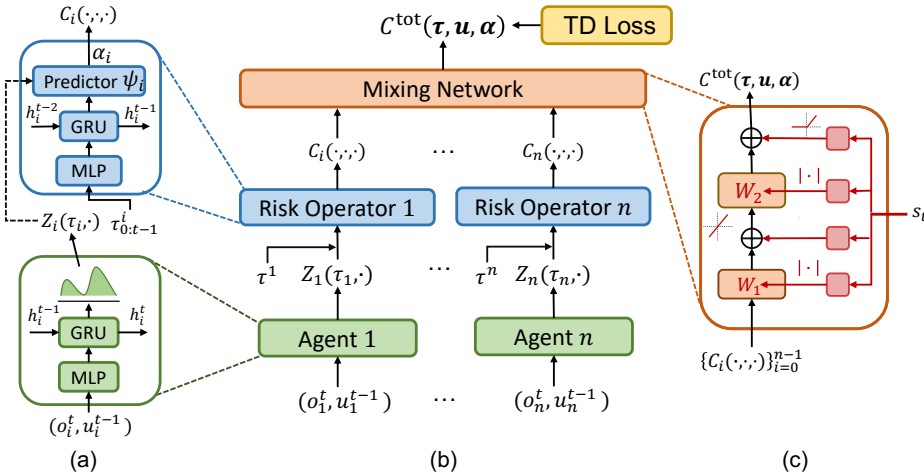

Figure 2: The framework of RMIX (dotted arrow indicates that gradients are blocked during training). (a) Agent network structure (bottom) and risk operator (top). (b) The overall architecture. (c) Mixing network structure. Each agent $i$ applies an individual risk operator $\Pi_{\alpha_i}$ on its return distribution $Z_i(\cdot, \cdot)$ to calculate $C_i(\cdot, \cdot, \cdot)$ for execution given risk level $\alpha_i$ predicted by the dynamic risk level predictor $\psi_i$. $\{C_i(\cdot, \cdot, \cdot)\}_{i=0}^{n-1}$ are fed into the mixing network for centralized training.

of Dirac Delta ($\delta$) functions [3], which is demonstrated to be highly expressive and computationally efficient (Bellemare et al., 2017). For convenience, we provide the definition of the *Generalized Return Probability Density Function (PDF)*.

**Definition 1.** *(Generalized Return PDF). For a discrete random variable $R \in [-R_{\max}, R_{\max}]$ and probability mass function (PMF) $\mathscr{P}(R = r_k)$, where $r_k \in [-R_{\max}, R_{\max}]$, we define the generalized return PDF as: $f_R(r) = \sum_{r_k \in R} \mathscr{P}_R(r_k)\delta(r - r_k)$. Note that for any $r_k \in R$, the probability of $R = r_k$ is given by the coefficient of the corresponding $\delta$ function, $\delta(r - r_k)$.*

We define the return distribution of each agent $i$ at time step $t$ as:

$$Z_i^t(\tau_i, u_i^{t-1}) = \sum_{j=1}^m \mathscr{P}_j(\tau_i, u_i^{t-1})\delta_j(\tau_i, u_i^{t-1}) \tag{1}$$

where $m$ is the number of Dirac Delta functions. $\delta_j(\tau_i, u_i^{t-1})$ is the $j$-th Dirac Delta function and indicates the estimated value which can be parameterized by neural networks in practice. $\mathscr{P}_j(\tau_i, u_i^{t-1})$ is the corresponding probability of the estimated value given local observations and actions. $\tau_i$ and $u_i^{t-1}$ are trajectories (up to that timestep) and actions of agent $i$, respectively. With the individual return distribution $Z_i^t(\tau_i, u_i^{t-1}) \in \mathcal{Z}$ and cumulative distribution function (CDF) $F_{Z_i(\tau_i, u_i^{t-1})}$, we define the CVaR operator $\Pi_{\alpha_i}$, at a risk level $\alpha_i$ ($\alpha_i \in (0, 1]$ and $i \in \mathcal{A}$) over return as[4]

$$C_i^t(\tau_i, u_i^{t-1}, \alpha_i) = \Pi_{\alpha_i^t} Z_i^t(\tau_i, u_i^{t-1}) = \mathrm{CVaR}_{\alpha_i^t}(F_{Z_i^t(\tau_i, u_i^{t-1})}), \tag{2}$$

where $C \in \mathcal{C}$. As we use CVaR on return distributions, it corresponds to risk-neutrality (expectation, $\alpha_i = 1$) and indicates the improving degree of risk-aversion ($\alpha_i \to 0$). $\mathrm{CVaR}_{\alpha_i}$ can be estimated in a nonparametric way given ordering of Dirac Delta functions $\{\delta_j\}_{j=1}^m$ (Kolla et al., 2019) by leveraging the individual distribution:

$$\mathrm{CVaR}_{\alpha_i} = \sum_{j=1}^m \mathscr{P}_j \delta_j \mathbf{1}\{\delta_j \leq \hat{v}_{m,\alpha_i}\}, \tag{3}$$

where $\mathbf{1}\{\cdot\}$ is the indicator function and $\hat{v}_{m,\alpha_i}$ is estimated value at risk from $\hat{v}_{m,\alpha_i} = \lfloor \delta_{m(1-\alpha_i)} \rfloor$ with $\lfloor \cdot \rfloor$ being floor function. This is a closed-form formulation and can be easily implemented in practice. The optimal action of agent $i$ can be calculated via $\arg\max_{u_i} C_i(\tau_i, u_i^{t-1}, \alpha_i)$. We will introduce the decentralized execution in detail in Section 3.3.

---

[3]The Dirac Delta is a *Generalized function* in the theory of distributions and not a function given the properties of it, we use the name *Dirac Delta function* by convention.

[4]We will omit $t$ in the rest of the paper for notation brevity.

## 3.2 DYNAMIC RISK LEVEL PREDICTION

The values of risk levels, i.e., $\alpha_i$, $i \in \mathcal{A}$, are important for the agents to make decisions. Most of the previous works take a fixed value of risk level and do not take into account any temporal structure of agents' trajectories, which can impede centralized training in the evolving multi-agent environments. Therefore, we propose the dynamic risk level prediction, which determines the risk levels of agents by explicitly taking into account the temporal nature of the stochastic outcomes, to alleviate time-consistency issue (Ruszczyński, 2010; Iancu et al., 2015) and stabilize the centralized training. Specifically, we represent the risk operator $\Pi_\alpha$ by a deep neural network, which calculates the CVaR value with predicted dynamic risk level $\alpha$ over the return distribution.

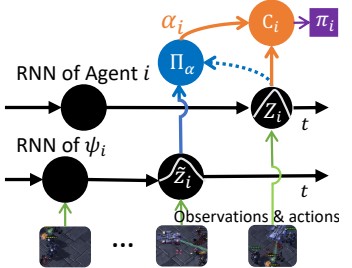

Figure 3: Agent architecture.

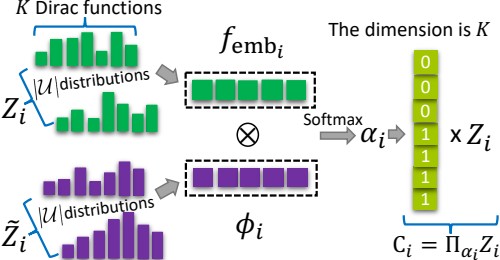

Figure 4: Risk level predictor $\psi_i$.

We show the architecture of agent $i$ in Figure 3 and illustrate how $\psi_i$ works with agent $i$ for CVaR calculation in practice in Figure 4. As depicted in Figure 4, at time step $t$, the agent's return distribution is $Z_i$ and its historical return distribution is $\tilde{Z}_i$. Then we conducts the inner product to measure the discrepancy between the embedding of individual return distribution $f_{\mathrm{emb}}(Z_i)$ and the embedding of past trajectory $\phi_i(\tau_i^{0:t-1}, u_i^{t-1})$ modeled by GRU (Chung et al., 2014). We discretize the risk level range into $K$ even ranges for the purpose of computing. The $k$-th dynamic risk level $\alpha_i^k$ is output from $\psi_i$ and the probability of $\alpha_i^k$ is defined as:

$$\mathscr{P}(\alpha_i^k) = \frac{\exp\left(\langle f_{\mathrm{emb}}(Z_i)^k, \phi_i^k \rangle\right)}{\sum_{k'=0}^{K-1} \exp\left(\langle f_{\mathrm{emb}}(Z_i)^{k'}, \phi_i^{k'} \rangle\right)}. \tag{4}$$

Then we get the $k \in [1, \ldots, K]$ with the maximal probability by $\arg\max$ and normalize it into $(0, 1]$, thus $\alpha_i = k/K$. The prediction risk level $\alpha_i$ is a scalar value and it is converted into to a $K$-dimensional mask vector where the first $\lfloor \alpha_i \times K \rfloor$ items are one and the rest items are zero. This mask vector is used to calculate the CVaR value (Eqn. 2 and 3) of each action-return distribution that contains $K$ Dirac functions. Finally, we obtain $C_i$ and the policy $\pi_i$ as illustrated in Figure 3. During training, $f_{\mathrm{emb}_i}$ updates its weights and the gradients of $f_{\mathrm{emb}_i}$ are blocked (the dotted arrow in Figure 3) in order to prevent changing the weights of the network of agent $i$.

We note that the predictor differs from the attention network used in previous works (Iqbal & Sha, 2019; Yang et al., 2020) because the agent's current return distribution and its return distribution of previous time step are separate inputs of their embeddings and there is no *key*, *query* and *value* weight matrices. The dynamic risk level predictors allow agents to determine the risk level dynamically based on historical return distributions.

## 3.3 RISK-SENSITIVE BELLMAN EQUATION

Motivated by the success of optimizing the CVaR value in single-agent RL (Chow & Ghavamzadeh, 2014), RMIX aims to maximize the CVaR value of the joint return distribution, rather than the expectation (Rashid et al., 2018; Son et al., 2019). As proved in Theorem 1, the maximizing operation of CVaR values satisfies the IGM principle, which implies that maximizing the CVaR value of joint return distribution is equivalent to maximizing the CVaR value of each agent.

**Theorem 1.** *In decentralized execution, given $\boldsymbol{\alpha} = \{\alpha_i\}_{i=0}^{n-1}$, we define the global $\arg\max$ performed on global CVaR $C^{\mathrm{tot}}(\boldsymbol{\tau}, \boldsymbol{u}, \boldsymbol{\alpha})$ as:*

$$\arg\max_{\boldsymbol{u}} C^{\mathrm{tot}}(\boldsymbol{\tau}, \boldsymbol{u}, \boldsymbol{\alpha}) = \left( \arg\max_{u_1} C_1(\tau_1, u_1, \alpha_1), \cdots, \arg\max_{u_n} C_n(\tau_n, u_n, \alpha_n) \right) \tag{5}$$

*where $\boldsymbol{\tau}$ and $\mathbf{u}$ are trajectories (up to that timestep) and actions of all agents, respectively. The individuals' maximization operation over return distributions defined above satisfies IGM and allows each agent to participate in a decentralised execution solely by choosing greedy actions with respect to its $C_i(\tau_i, u_i, \alpha_i)$.*

To maximize the CVaR value of each agent, we define the risk-sensitive Bellman operator $\mathcal{T}$:

$$\mathcal{T}C^{\text{tot}}(\boldsymbol{s}, \boldsymbol{u}, \boldsymbol{\alpha}) := \mathbb{E}\left[R(\boldsymbol{s}, \boldsymbol{u}) + \gamma \max_{\boldsymbol{u}'} C^{\text{tot}}(\boldsymbol{s}', \boldsymbol{u}', \boldsymbol{\alpha}')\right] \tag{6}$$

where $\boldsymbol{\alpha}$ and $\boldsymbol{\alpha}'$ are agents' static risk levels or dynamic risk levels output from the dynamic risk level predictor $\psi$ at each time step. The risk-sensitive Bellman operator $\mathcal{T}$ operates on the CVaR value of the agent and the reward, which can be proved to be a contracting operation, as showed in Proposition 1. Therefore, we can leverage the TD learning (Sutton & Barto, 2018) to compute the maximal CVaR value of each agent, thus leading to the maximal global CVaR value.

**Proposition 1.** $\mathcal{T} : \mathcal{C} \mapsto \mathcal{C}$ *is a $\gamma$-contraction.*

### 3.4 CENTRALIZED TRAINING

We introduce the centralized training of RMIX. We utilize the monotonic mixing network of QMIX, which is a value decomposition network via hypernetwork (Ha et al., 2017) to maintain the monotonicity and has shown success in cooperative MARL. Based on IGM (Son et al., 2019) principle, we define monotonic IGM between $C^{\text{tot}}$ and $C_i$ for RMIX:

$$\frac{\partial C^{\text{tot}}}{\partial C_i} \geq 0, \forall i \in \{1, 2, \ldots, N\}, \tag{7}$$

where $C^{\text{tot}}$ is the total CVaR value and $C_i(\tau_i, u_i)$ is the individual CVaR value of agent $i$, which can be considered as a latent combination of agents' implicit CVaR values to the global CVaR value. Following the CTDE paradigm, we define the TD loss of RMIX as:

$$\mathcal{L}_{\Pi}(\theta) := \mathbb{E}_{\mathcal{D}' \sim \mathcal{D}}\left[(y_t^{\text{tot}} - C^{\text{tot}}(\boldsymbol{s}_t, \boldsymbol{u}_t, \boldsymbol{\alpha}_t))^2\right] \tag{8}$$

where $y_t^{\text{tot}} = \left(r_t + \gamma \max_{\boldsymbol{u}'} C_{\bar{\theta}}^{\text{tot}}(\boldsymbol{s}_{t+1}, \boldsymbol{u}', \boldsymbol{\alpha}')\right)$, and $(y_t^{\text{tot}} - C_{\theta}^{\text{tot}}(\boldsymbol{s}_t, \boldsymbol{u}_t, \boldsymbol{\alpha}_t))$ is our CVaR TD error for updating CVaR values. $\theta$ is the parameters of $C^{\text{tot}}$ which can be modeled by a deep neural network and $\bar{\theta}$ indicates the parameters of the target network which is periodically copied from $\theta$ for stabilizing training (Mnih et al., 2015). While training, gradients from $Z_i$ are blocked to avoid changing the weights of the agents' network from the dynamic risk level predictor. We train RMIX in an end-to-end manner. $\psi_i$ is trained together the agent network via the loss defined in Eq. 8. During training, $f_{\text{emb}_i}$ updates its weights while gradients of $f_{\text{emb}_i}$ are blocked in order to prevent changing the weights of the return distribution in agent $i$. The pseudo code of RMIX is shown in Algorithm 1 in Appendix D. We present our framework as shown in Figure 2. Our framework is flexible and can be easily used in many cooperative MARL methods.

## 4 EXPERIMENTS

We empirically evaluate our methods on various StarCraft II scenarios. Especially, we are interested in the robust cooperation in complex *asymmetric* and *homogeneous/heterogeneous* scenarios. Additional introduction of baselines, scenarios and results are in Appendix C, E, F and G.

### 4.1 EXPERIMENT SETUP

**StarCraft II** We consider SMAC (Samvelyan et al., 2019) benchmark[5] (screenshots of some scenarios are in Figure 5), a challenging set of cooperative StarCraft II maps for micromanagement, as our evaluation environments. We evaluate our methods for every 10,000 training steps during training by running 32 episodes in which agents trained with our methods battle with built-in game bots. We report the mean test won rate (`test_battle_won_mean`, percentage of episodes won of MARL agents) along with one standard deviation of won rate (shaded in figures). Due to limited

---

[5]https://github.com/oxwhirl/smac

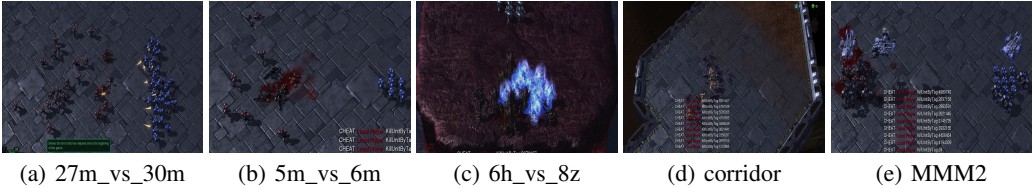

(a) 27m_vs_30m     (b) 5m_vs_6m     (c) 6h_vs_8z     (d) corridor     (e) MMM2

Figure 5: SMAC scenarios: 27m_vs_30m, 5m_vs_6m, 6h_vs_8z, corridor and MMM2.

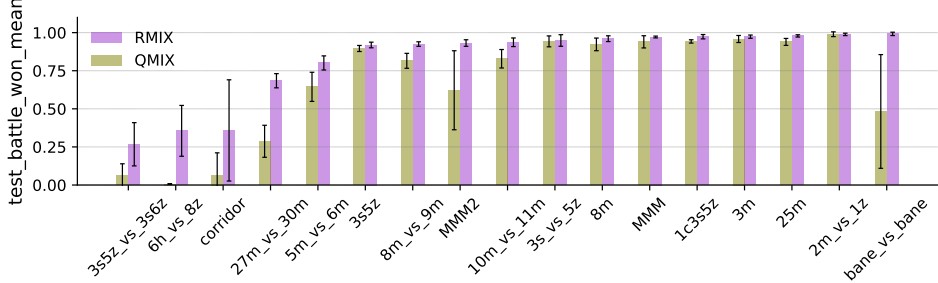

Figure 6: `test_battle_won_mean` summary of RMIX and QMIX on 17 SMAC scenarios.

page space, we present the results of our methods and baselines on 8 scenarios (we train our methods and baselines on 17 SMAC scenarios): corridor, 3s5z_vs_3s6z, 6h_vs_8z, 5m_vs_6m, 8m_vs_9m, 10m_vs_11m, 27m_vs_30m and MMM2. Table 1 shows detailed information on these scenarios.

**Baselines and training** The baselines are IQL (Tampuu et al., 2017), VDN (Sunehag et al., 2017), COMA (Foerster et al., 2017a), QMIX (Rashid et al., 2018), QTRAN (Son et al., 2019), MAVEN (Mahajan et al., 2019) and Qatten (Yang et al., 2020). We implement our methods on PyMARL[6] and use 5 random seeds for training each method on 17 SMAC scenarios. We carry out experiments on NVIDIA Tesla V100 GPU 16G.

## 4.2 EXPERIMENT RESULTS

RMIX demonstrates substantial superiority over baselines in *asymmetric* and *homogeneous* scenarios as depicted in Figure 7. RMIX outperforms baselines in *asymmetric homogeneous* scenarios: 5m_vs_6m, 8m_vs_9m, 10m_vs_11m and 27m_vs_30m (**hard** game). In 3s5z_vs_3s6z (*asymmetric heterogeneous*, **very hard** game) and MMM2 (*symmetric heterogeneous*, **hard** game), RMIX also shows leading performance over baselines. RMIX learns micro-trick (wall off) and micro-trick (focus fire) faster and better in **very hard** corridor and 6h_vs_8z. RMIX improves coordination in a sample efficient way via risk-sensitive policies. We summarize the performance of

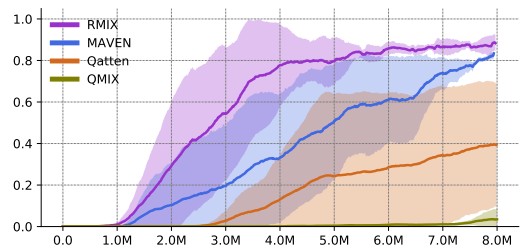

Figure 8: `test_battle_won_mean` of RMIX, QMIX, Qatten and MAVEN on **very hard** corridor scenario.

RMIX and QMIX on 17 SMAC scenarios in Figure 6. Readers can refer to Figure 19 and 20 for more results. We present results of RMIX in Figure 13 and 14 on 3s5z_vs_3s6z and 6h_vs_8z in 8 million training steps. Although training 27m_vs_30m is memory-consuming, we also present results of 2.5 million training steps, as depicted in Figure 15.

Interestingly, as illustrated in Figure 8, RMIX also demonstrates leading exploration performance over baselines on **very hard** corridor (in Figure 5(d)) scenario, where there is a narrow corridor con-

---

[6]`https://github.com/oxwhirl/pymarl`

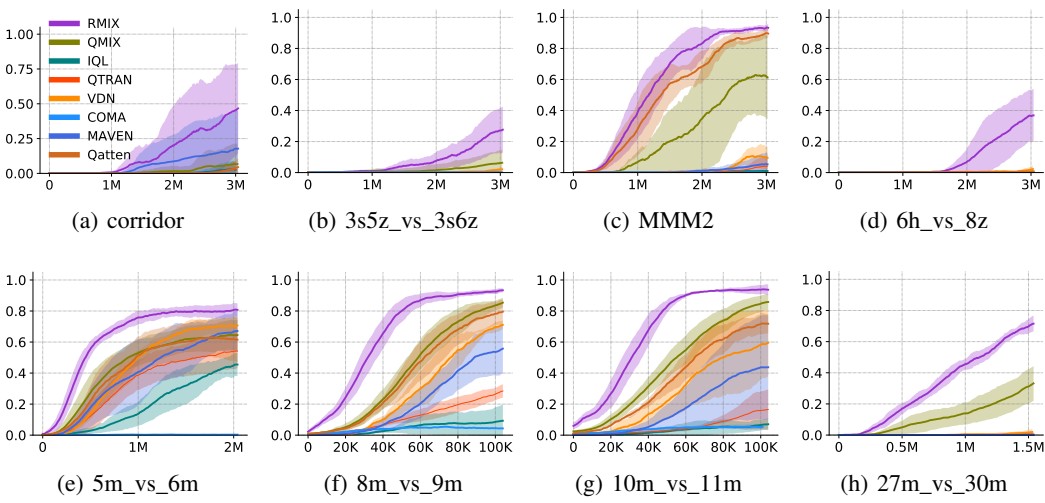

Figure 7: `test_battle_won_mean` of RMIX and baselines on 8 scenarios, the x-axis denotes the training steps and the y-axis is the test battle won rate, ranging from 0 to 1. It applies for result figures in the rest of the paper, including figures in appendix.

necting two separate rooms, and agents should learn to cooperatively combat the opponents to avoid being beaten by opponents with the divide-and-conquer strategy. RMIX outperforms MAVEN (Mahajan et al., 2019), which was originally proposed for tackling multi-agent exploration problems, both in sample efficiency and final results. After 4 million training steps, RMIX starts to converge while MAVEN starts to converge after over 7 million training steps.

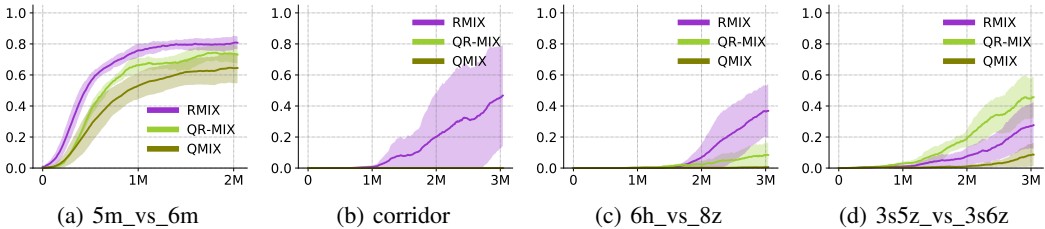

Figure 9: `test_battle_won_mean` of RMIX vs QR-MIX and QMIX.

In addition, we compare RMIX with QR-MIX (Hu et al., 2020). We implement QR-MIX with PyMARL and train it on 5m_vs_6m (hard), corridor (very hard), 6h_vs_8z (very hard) and 3s5z_vs_3s6z (very hard) with 3 random seeds for each scenario. Hyper-parameters used during training are from QR-MIX paper. As shown in Figure 9, RMIX shows leading performance and superior sample efficiency over QR-MIX on 5m_vs_6m , 27m_vs_30m and 6h_vs_8z. With distributional RL, QR-MIX presents slightly better performance on 3s5z_vs_3s6z. We present more results of RMIX vs QR-MIX in Appendix G.3.

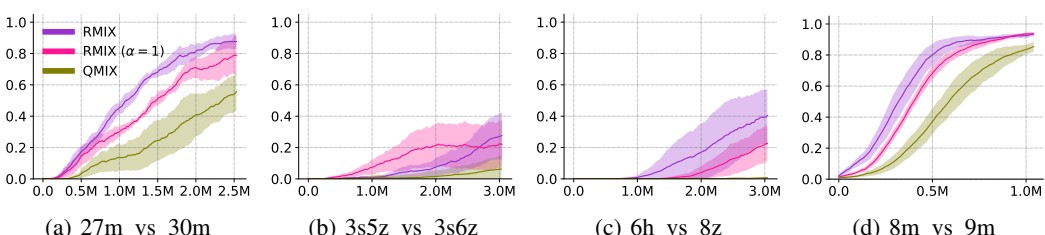

Figure 10: `test_battle_won_mean` of RMIX and baselines on 4 scenarios.

We conduct an ablation study by fixing the risk level with the value of 1, thus we get the risk-neutral method, which we name as RMIX ($\alpha = 1$). We present results of RMIX, RMIX ($\alpha = 1$) and QMIX

on 4 scenarios in Figure 10. RMIX outperforms RMIX ($\alpha = 1$) in many *heterogeneous* and *asymmetric* scenarios, demonstrating the benefits of learning risk-sensitive MARL policies in complex scenarios where the potential of loss should be taken into consideration in coordination. Intuitively, for *asymmetric* scenarios, agents can be easily defeated by the opponents. As a consequence, coordination between agents is cautious in order to win the game, and the cooperative strategies in these scenarios should avoid massive casualties in the starting stage of the game. Apparently, our risk-sensitive policy representation works better than vanilla expected Q values in evaluation. In *heterogeneous* scenarios, action space and observation space are different among different types of agents, and methods with vanilla expected action value are inferior to RMIX.

To show that our proposed method is flexible in other mixing network for the ablation study, we apply additivity of individual CVaR values to represent the global CVaR value as $C^{\text{tot}}(\boldsymbol{\tau}, \boldsymbol{u}, \boldsymbol{\alpha}) = C_1(\tau_1, u_1, \alpha_1) + \cdots + C_n(\tau_n, u_n, \alpha_n)$. Following the training of RMIX, we name this method Risk Decomposition Network (RDN) for ablation study. we use experiment setup of VDN and train RDN on 5 SMAC scenarios. With CVaR values, RDN outperforms VDN and QMIX in 1c3s5z, 5m_vs_6m, 8m_vs_9m and MMM2 with convincing improvements, as depicted in Figure 11. In some scenar-

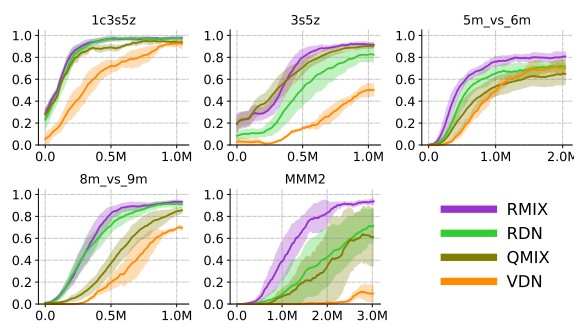

Figure 11: `test_battle_won_mean` of RMIX, RDN, QMIX and VDN on 5 SMAC scenarios.

ios, for example 1c3z5z and 8m_vs_9m, the converged performance is even equal to that of RMIX, which demonstrate that RMIX is flexible in additivity mixing networks. Overall, with the new policy representation and additivity decomposition network, we can gain convincing improvements of RDN over VDN.

We present how the risk level $\alpha$ of each agent changes during the episode and emergent cooperative strategies between agents in the results analysis of RMIX in Appendix G.2 due to limited space.

## 5 CONCLUSION

In this paper, we propose RMIX, a novel risk-sensitive MARL method with CVaR over the learned distributions of individuals' Q values. Our main contributions are in three folds: (i) We first learn the return distributions of individuals to analytically calculate CVaR for decentralized execution; (ii) We then propose a dynamic risk level predictor for CVaR calculation to handle the temporal nature of the stochastic outcomes during executions; (iii) We finally propose risk-sensitive Bellman equation along with *Individual-Global-MAX* (IGM) for MARL training. Empirically, we show that RMIX significantly outperforms state-of-the-art methods on many challenging StarCraft II tasks, demonstrating enhanced coordination in many complex scenarios and revealing high sample efficiency. To the best of our knowledge, our work is the *first* attempt to investigate cooperative MARL with risk-sensitive policies under the Dec-POMDP framework.

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

## A    RELATED WORKS

As deep reinforcement learning (DRL) becomes prevailing (Mnih et al., 2015; Schulman et al., 2017), recent years have witnessed a renaissance in cooperative MARL with deep learning. However, there are several inevitable issues, including the nonstationarity of the environment from the view of individual agents (Foerster et al., 2017b), the credit assignment in cooperative scenarios with shared global rewards (Sunehag et al., 2017; Rashid et al., 2018), the lack of coordination and communication in cooperative scenarios (Jiang & Lu, 2018; Wang et al., 2020b) and the failure to consider opponents' strategies when learning agent policies (He et al., 2016). Aiming to address these issues, *centralized training with decentralized execution* (CTDE) (Oliehoek et al., 2008; Kraemer & Banerjee, 2016) has drawn enormous attention via training policies of each agent with access to global trajectories in a centralized way and executing actions given only the local observations of each agent in a decentralized way. Several MARL methods are proposed (Lowe et al., 2017; Foerster et al., 2017a; Sunehag et al., 2017; Rashid et al., 2018; Son et al., 2019; Yang et al., 2020; Wang et al., 2020a), including value-based and policy gradient. Among these methods, VDN (Sunehag et al., 2017) and QMIX (Rashid et al., 2018) are representative methods that use value decomposition of the joint action-value function by adopting additivity and monotonicity structural constraints. Free from such structural constraints, QTRAN (Son et al., 2019) guarantees more general factorization than VDN and QMIX, however, such linear affine transformation fails to scale up in complex multi-agent scenarios, for example, StarCraft II environments.

However, most of works focus on decomposing the global Q value into individual Q values with different formulas and network architectures, either with structural constraints, for example additivity and monotonicity (Sunehag et al., 2017; Rashid et al., 2018), or with estimating values that forms a new global Q value estimation, for example representing Q values via summation of state-value and advantage or multi-head attention networks (Wang et al., 2020a; Yang et al., 2020). Current MARL methods neglect the limited representation of agent values, which fails to consider the agent-level impact of individuals to the whole system when transforming individual utilities to global values, leading to hostile training in complex scenarios. Typically, the problem of random cost underlying the nonstationarity of the environment, a.k.a risk-sensitive learning, which is very important for many real-world applications, has rarely been investigated. Current work are either confined in simple settings (Reddy et al., 2019) or have no general framework and convincing results in complex domains (Lyu & Amato, 2020). Further research should be done in risk-sensitive cooperative MARL. We propose RMIX and fill the gap in this field.

Recent advances in distributional RL (Bellemare et al., 2017; Dabney et al., 2018a;b) focuses on learning distribution over returns. However, with return distributions, these works still focus on either risk-neutral settings or with static risk level in single-agent setting, which neglects the ubiquitous significant risk-sensitive problems in many multi-agent real-world applications, including pipeline robots cooperation in factories, warehouse robots coordination, etc. This is very common in real-world applications, especially in highly dynamic tasks for example military action, resource allocation, finance portfolio management and Internet of Things (IoT), etc.

Chow & Ghavamzadeh (2014) proposed considered the mean-CVaR optimization problem in MDPs and proposed policy gradient with CVaR, and García et al. (2015) presented a survey on safe RL, which have ignited the research on borrowing risk measures in RL (García et al., 2015; Tamar et al., 2015; Tang et al., 2019; Hiraoka et al., 2019; Majumdar & Pavone, 2020; Keramati et al., 2020; Ma et al., 2020). However, these works focus on single-agent settings, where there is one agent interacting with the environment compared with dynamic and non-stationary multi-agent environments. The merit of CVaR in optimization of MARL has yet to be discovered. We explore the CVaR risk measure in our methods and demonstrate the leading performance of our MARL methods.

## B    PROOFS

We present proofs on our propositions introduced in previous sections. The proposition and equations numbers are reused in restated propositions.

**Assumption 1.** *The mean rewards are bounded in a known interval, i.e., $r \in [-R_{\max}, R_{\max}]$.*

This assumption means we can bound the absolute value of the Q-values as $|Q_{sa}| \leq Q_{\max} = HR_{\max}$, where $H$ is the maximum time horizon length in episodic tasks.

**Proposition 1.** $\mathcal{T} : \mathcal{C} \mapsto \mathcal{C}$ *is a $\gamma$-contraction.*

*Proof.* We consider the sup-norm contraction,

$$\left| \mathcal{T} C_{(1)}(s, u, \alpha_{(1)}) - \mathcal{T} C_{(2)}(s, u, \alpha_{(2)}) \right| \leq \gamma \left\| C_{(1)}(s, u, \alpha_{(1)}) - C_{(2)}(s, u, \alpha_{(2)}) \right\|_\infty \tag{9}$$
$$\forall s \in \mathcal{S}, u \in \mathcal{U}, \alpha_{(i), i \in \{1,2\}} \in \mathcal{A}.$$

The sup-norm is defined as $\|C\|_\infty = \sup_{s \in \mathcal{S}, u \in \mathcal{U}, \alpha \in \mathcal{A}} |C(s, u)|$ and $C \in \mathbb{R}$.

In $\{C_i\}_{i=0}^{n-1}$, the risk level is fixed and can be considered implicit input. Given two risk level set $\alpha_{(1)}$ and $\alpha_{(2)}$, and two different return distributions $Z_{(1)}$ and $Z_{(2)}$, we prove:

$$\begin{aligned}
&\left| \mathcal{T} C_{(1)} - \mathcal{T} C_{(2)} \right| \\
&\leq \max_{s, u} \left| \left[ \mathcal{T} C_{(1)} \right] (s, u, \alpha_{(1)}) - \left[ \mathcal{T} C_{(2)} \right] (s, u, \alpha_{(1)}) \right| \\
&= \max_{s, u} \left| \gamma \sum_{s'} \mathcal{P}(s'|s, u) \left( \max_{u'} C_{(1)}(s', u', \alpha') - \max_{u'} C_{(2)}(s', u', \alpha') \right) \right| \\
&\leq \gamma \max_{s'} \left| \max_{u'} C_{(1)}(s', u', \alpha') - \max_{u'} C_{(2)}(s', u', \alpha') \right| \\
&\leq \gamma \max_{s', u'} \left| C_{(1)}(s', u', \alpha') - C_{(2)}(s', u', \alpha') \right| \\
&= \gamma \left\| C_{(1)} - C_{(2)} \right\|_\infty
\end{aligned} \tag{10}$$

This further implies that

$$\left| \mathcal{T} C_{(1)} - \mathcal{T} C_{(2)} \right| \leq \gamma \left\| C_{(1)} - C_{(2)} \right\|_\infty \quad \forall s \in \mathcal{S}, u \in \mathcal{U}, \alpha_{(i), i \in \{1,2\}} \in \mathcal{A}. \tag{11}$$
$\square$

With proposition 1, we can leverage the TD learning (Sutton & Barto, 2018) to compute the maximal CVaR value of each agent, thus leading to the maximal global CVaR value. In some scenarios, where risk is not the primal concern for policy optimization, for example corridor scenario, where agents should learn to explore to win the game. RMIX will show less sample efficient to learn the optimal policy compared with its risk neutral variant, RMIX ($\alpha = 1$). As we can see in Figure 16(b), section G.1.1, RMIX learns slower than RMIX ($\alpha = 1$) because it relies on the dynamic risk predictor while the predictor is trained together with the agent, it will take more samples in these environments. Interestingly, RMIX shows very good performance over other baselines.

**Theorem 1.** *In decentralized execution, given $\alpha = \{\alpha_i\}_{i=0}^{n-1}$, we define the global $\arg\max$ performed on global CVaR $C^{\text{tot}}(\tau, u, \alpha)$ as:*

$$\arg\max_u C^{\text{tot}}(\tau, u, \alpha) = \left( \arg\max_{u_1} C_1(\tau_1, u_1, \alpha_1), \cdots, \arg\max_{u_n} C_n(\tau_n, u_n, \alpha_n) \right) \tag{5}$$

*where $\tau$ and $\mathbf{u}$ are trajectories (up to that timestep) and actions of all agents, respectively. The individuals' maximization operation over return distributions defined above satisfies IGM and allows each agent to participate in a decentralised execution solely by choosing greedy actions with respect to its $C_i(\tau_i, u_i, \alpha_i)$.*

*Proof.* With monotonicity network $f_{\mathrm{m}}$, in RMIX, we have

$$C^{\text{tot}}(\tau, u, \alpha) = f_{\mathrm{m}}(C_1(\tau_1, u_1, \alpha_1), \ldots, C_n(\tau_n, u_n, \alpha_n)) \tag{12}$$

Consequently, we have

$$C^{\text{tot}}(\tau, \{\arg\max_{u'} C_i(\tau_i, u', \alpha_i)\}_{i=0}^{n-1}) = f_{\mathrm{m}}(\{\max_{u'} C_i(\tau_i, u', \alpha_i)\}_{i=0}^{n-1}) \tag{13}$$

By the monotonocity property of $f_{\mathrm{m}}$, we can easily derive that if $j \in \{0, 1, \ldots, n-1\}$, $u_j^* = \arg\max_{u'} C_j(\tau_j, u', \alpha_j)$, $\alpha_j^* \in (0, 1]$ is the optimal risk level given the current return distributions and historical return distributions, and actions of other agents are not the best action, then we have

$$f_{\mathrm{m}}(\{C_j(\tau_j, u_j, \alpha_j)\}_{i=0}^{n-1}) \leq f_{\mathrm{m}}(\{C_j(\tau_j, u_j, \alpha_j)\}_{i=0, i \neq j}^{n-1}, C_j(\tau_j, u_j^*, \alpha_j)). \tag{14}$$

So, for all agents, $\forall j \in \{0, 1, \ldots, n-1\}$, $u_j^* = \arg\max_{u'} C_j(\tau_j, u', \alpha_j)$, we have

$$
\begin{aligned}
f_{\mathrm{m}}(\{C_j(\tau_i, u_i, \alpha_i)\}_{i=0}^{n-1}) &\leq f_{\mathrm{m}}(\{C_j(\tau_j, u_j, \alpha_j)\}_{i=0, i\neq j}^{n-1}, C_j(\tau_j, u_j^*)) \\
&\leq f_{\mathrm{m}}(\{C_i(\tau_i, u_i^*, \alpha_i)\}_{i=0}^{n-1}) \\
&= \max_{\{u_i\}_{i=0}^{n-1}} f_{\mathrm{m}}(\{C_i(\tau_i, u_i, \alpha_i)\}_{i=0}^{n-1}).
\end{aligned}
\tag{15}
$$

Therefore, we can get

$$
\max_{\{u_i, \alpha_i\}_{i=0}^{n-1}} f_{\mathrm{m}}(\{C_i(\tau_i, u_i, \alpha_i)\}_{i=0}^{n-1}) = \max_{\boldsymbol{u}, \alpha} C^{\mathrm{tot}}(\boldsymbol{\tau}, \boldsymbol{u}, \boldsymbol{\alpha}),
\tag{16}
$$

which implies

$$
\max_{\boldsymbol{u}} C^{\mathrm{tot}}(\boldsymbol{\tau}, \boldsymbol{u}, \boldsymbol{\alpha}) = C^{\mathrm{tot}}(\boldsymbol{\tau}, \{\arg\max_{u', \alpha'} C_i(\tau_i, u', \alpha_i)\}_{i=0}^{n-1}).
\tag{17}
$$

$\square$

**Proposition 2.** *For any agent $i$, $i \in \{0, \ldots, n-1\}$, $\exists \lambda(\tau_i, u_i) \in (0, 1]$, such that $C_i(\tau_i, u_i) = \lambda(\tau_i, u_i) \mathbb{E}\left[Z_i(\tau_i, u_i)\right]$.*

*Proof.* We first provide that given a return distribution $Z$, return random variable $\mathscr{Z}$ and risk level $\alpha \in \mathcal{A}$, $\forall z$, $\Pi_\alpha Z$ can be rewritten as $\mathbb{E}\left[\mathscr{Z} | \mathscr{Z} < z\right] < \mathbb{E}\left[\mathscr{Z}\right]$. This can be easily proved by following Privault (2020)'s proof. Thus we can get $\Pi_\alpha Z < \mathbb{E}[Z]$, and there exists $\lambda_{(\tau_i, u_i)} \in (0, 1]$, which is a value of agent's trajectories, such that $\Pi_\alpha Z_i(\tau_i, u_i) = \lambda_{(\tau_i, u_i)} \mathbb{E}\left[Z_i(\tau_i, u_i)\right]$. $\square$

Proposition 2 implies that we can view the CVaR value as truncated values of Q values that are in the lower region of return distribution $Z_i(\tau_i, u_i)$. CVaR can be decomposed into two factors: $\lambda_{(\tau_i, u_i)}$ and $\mathbb{E}[Z_i(\tau_i, u_i)]$.

## C  ADDITIONAL BACKGROUND

We introduce additional background on cooperative MARL algorithms, including QMIX, MAVEN and Qatten, for the convenience of readers who want to know more on these algorithms.

Q-based cooperative MARL is concerned with estimating an accurate action-value function to select the actions with maximum expected returns. The optimal Q-function is defined as (Rashid et al., 2018):

$$
\begin{aligned}
Q_\theta^{\mathrm{tot}}(\boldsymbol{s}, \mathbf{u}) &:= \mathbb{E}\left[\sum_{t=0}^{\infty} \gamma^t r(\boldsymbol{s}_t, \mathbf{u}_t, \theta) \mid \begin{array}{c} \boldsymbol{s}_{t+1} \sim \mathscr{P}(\cdot \mid \boldsymbol{s}_t, \mathbf{u}_t, \theta), \\ \mathbf{u}_{t+1} = \arg\max Q_\theta^{\mathrm{tot}}(\boldsymbol{s}_{t+1}, \cdot) \end{array}\right] \\
&= r(\boldsymbol{s}, \mathbf{u}, \theta) + \gamma \mathbb{E}\left[\max Q_\theta^{\mathrm{tot}}(\boldsymbol{s}', \cdot) \mid \boldsymbol{s}' \sim \mathscr{P}(\cdot \mid \boldsymbol{s}, \mathbf{u}, \theta)\right],
\end{aligned}
$$

where $\theta$ is the parameters of $Q^{\mathrm{tot}}$ which can be model by deep neural networks working as the Q-function approximator. This Q-network can be trained by minimizing the loss function in a supervised-learning fashion as defined below:

$$
\mathcal{L}(\theta) := \mathbb{E}_{\mathcal{D}' \sim \mathcal{D}}\left[\underbrace{\left(r_t + \gamma Q_{\bar{\theta}}^{\mathrm{tot}}\left(\boldsymbol{s}_{t+1}, \arg\max_{\boldsymbol{u}} Q_\theta^{\mathrm{tot}}(\boldsymbol{s}_{t+1}, \cdot)\right)}_{y_t^{\mathrm{tot}}} - Q_\theta^{\mathrm{tot}}(\boldsymbol{s}_t, \boldsymbol{u}_t))^2\right],
$$

where $\mathcal{D}' = (\boldsymbol{s}_t, \mathbf{u}_t, r_t, \boldsymbol{s}_{t+1})$ and $\mathcal{D}$ is the replay buffer. $\bar{\theta}$ indicates the parameters of the target network which is periodically copied from $\theta$ for stabilizing training (Mnih et al., 2015). The network is trained in a centralized way with all partial observations accessible to all agents.

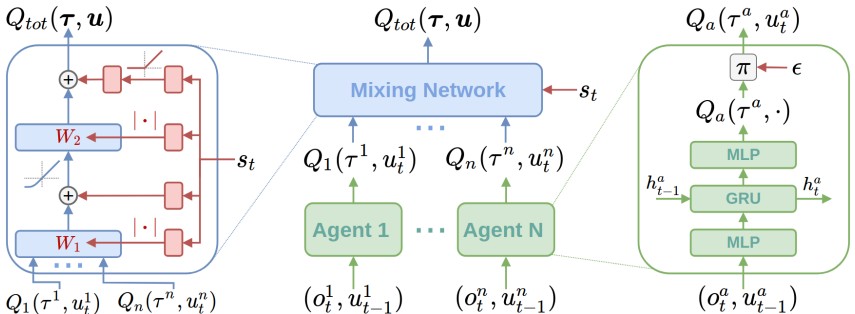

Figure 12: The overall setup of QMIX (best viewed in colour), reproduced from the original paper (Rashid et al., 2018) (a) Mixing network structure. In red are the hypernetworks that produce the weights and biases for mixing network layers shown in blue. (b) The overall QMIX architecture. (c) Agent network structure.

**QMIX** QMIX (Rashid et al., 2018) is a well-known multi-agent Q-learning algorithm in the centralised training and decentralised execution paradigm, which restricts the joint action Q-values it can represent to be a monotonic mixing of each agent's utilities in order to enable decentralisation and value decomposition:

$$\mathcal{Q}^{\text{mix}} := \left\{ Q_{tot} \mid Q_{tot}(\boldsymbol{\tau}, \mathbf{u}) = f_{\text{m}} \left( Q^1 \left( \tau^1, u^1 \right), \ldots Q_n \left( \tau^n, u^n \right) \right), \frac{\partial f_{\text{m}}}{\partial Q_a} \geq 0, Q_a(\tau, u) \in \mathbb{R} \right\}$$

and the $\arg\max$ operator is used to get the $Q_{tot}$ for centralized training via TD loss similar to DQN (Mnih et al., 2015)

$$\arg\max_{\mathbf{u}} Q_{tot}(\boldsymbol{\tau}, \mathbf{u}) = \begin{pmatrix} \arg\max_{u^1} Q_1(\tau^1, u^1) \\ \vdots \\ \arg\max_{u^n} Q_n(\tau^n, u^n) \end{pmatrix}.$$

The architecture is shown in Figure 12. The monotonic mixing network $f_{\text{m}}$ is parametrised as a feedforward network, of which non-negative weights are generated by hypernetworks (Ha et al., 2017) with the state as input.

**MAVEN** MAVEN (Mahajan et al., 2019) (multi-agent variational exploration) overcomes the detrimental effects of QMIX's monotonicity constraint on exploration via learning a diverse ensemble of monotonic approximations with the help of a latent space. It consists of value-based agents that condition their behaviour on the shared latent variable $z$ controlled by a hierarchical policy that offloads $\epsilon$-greedy with committed exploration. Thus, fixing $z$, each joint action-value function is a monotonic approximation to the optimal action-value function that is learnt with Q-learning.

**Qatten** Qatten (Yang et al., 2020) explicitly considers the agent-level impact of individuals to the whole system when transforming individual $Q^i$s into $Q_{\text{tot}}$. It theoretically derives a general formula of $Q_{\text{tot}}$ in terms of $Q^i$, based on a multi-head attention formation to approximate $Q_{\text{tot}}$, resulting in not only a refined representation of $Q_{\text{tot}}$ with an agent-level attention mechanism, but also a tractable maximization algorithm of decentralized policies.

# D  PSEUDO CODE OF RMIX

---

**Algorithm 1:** RMIX

---

**input:** $K, \gamma$;

1  Initialize parameters $\theta$ of the network of agent, risk operator and monotonic mixing network;
2  Initialize parameters $\bar{\theta}$ of the target network of agent, risk operator and monotonic mixing network;
3  Initialize replay buffer $\mathcal{D}$;
4  **for** $e \leftarrow 0$ **to** *MAX_EPISODE* **do**
5     Start a new episode;
6     **while** *EPISODE_IS_NOT_TEMINATED* **do**
7        Get the global state $\boldsymbol{s}^t$;
8        **for** *agent* $i \leftarrow 0$ **to** $n-1$ **do**
9           Get observation $o_i^t$ from the environment;
10          Get action of last step $u_i^{t-1}$ from the environment;
11          Estimate the local return distribution $Z_i^t(o_i^t, u_i^{t-1})$;
12          Predict the risk level $\alpha_i$:

$$\arg\max_k \left\{ \frac{\exp\big(\langle f_{\mathrm{emb}}(Z_i)^k, \phi_i^k \rangle\big)}{\sum_{k'=0}^{K-1} \exp\big(\langle f_{\mathrm{emb}}(Z_i)^{k'}, \phi_i^{k'} \rangle\big)} \right\}_k / K, k \in \{1, \ldots, K\};$$

13          Calculate CVaR values $C_i^t\left(o_i^t, u_i^{t-1}, \alpha_i\right) = \Pi_{\alpha_i^t} Z_i^t\left((o_i^t, u_i^{t-1})\right)$;
14          Get the action $u_i^t = \arg\max_{u^t} C_i^t\left(o_i^t, u_i^{t-1}, \alpha_i\right)$;
15       Concatenate $u_i^t, i \in [0, .., n-1]$ into $\mathbf{u}_t$;
16       Execute $\boldsymbol{u}_i^t$ into environment;
17       Receive global reward $r^t$ and observe a new state $\boldsymbol{s}'$;
18       Store $(\boldsymbol{s}^t, \{o_i^t\}_{i \in [0, \ldots, n-1]}, \boldsymbol{u}^t, r^t, \boldsymbol{s}')$ in replay buffer $\mathcal{D}$;
19       **if** *UPDATE* **then**
20          Sample a min-batch $\mathcal{D}'$ from replay buffer $\mathcal{D}$;
21          For each sample in $\mathcal{D}'$, calculate CVaR value $C_i$ by following steps in line 9-13;
22          Concatenate CVaR values $\{[C_0^0, \ldots, C_{n-1}^0]_0, \ldots, [C_0^{|\mathcal{D}'|-1}, \ldots, C_{n-1}^{|\mathcal{D}'|-1}]_{|\mathcal{D}'|-1}\}$;
23          For each $[C_0^j, \ldots, C_{n-1}^j]_{0, j \in [0, \ldots, |\mathcal{D}'|-1]}$, calculate $C_j^{\mathrm{tot}}$ in the mixing network;
24          Calculate the target value $y^{\mathrm{tot}} = \left(r_t + \gamma \max_{\boldsymbol{u}'} C_{\bar{\theta}}^{\mathrm{tot}}\right)$;
25          Calculate the TD loss $\mathcal{L}_\Pi(\theta) := \mathbb{E}_{\mathcal{D}' \sim \mathcal{D}}\left[(y^{\mathrm{tot}} - C^{\mathrm{tot}})^2\right]$;
26          Update $\theta$ by minimizing the TD loss;
27          Update $\bar{\theta}$: $\bar{\theta} \leftarrow \theta$;
28 **return** $\theta$;

---

# E    ADDITIONAL ENVIRONMENT INTRODUCTION

SMAC benchmark is a challenging set of cooperative StarCraft II maps for micromanagement developed by Samvelyan et al. (2019) built on DeepMind's PySC2 (Vinyals et al., 2017). We introduce **states and observations**, **action space** and **rewards** of SMAC, and **environmental settings of RMIX** below.

**States and Observations** At each time step, agents receive local observations within their field of view. This encompasses information about the map within a circular area around each unit with a radius equal to the sight range. The sight range makes the environment partially observable for each agent. An agent can only observe other agents if they are both alive and located within its sight range. Hence, there is no way for agents to distinguish whether their teammates are far away or dead. The feature vector observed by each agent contains the following attributes for both allied and enemy units within the sight range: distance, relative x, relative y, health, shield, and unit type. All Protos units have shields, which serve as a source of protection to offset damage and can regenerate if no new damage is received. The global state is composed of the joint observations but removing the restriction of sight range, which could be obtained during training in the simulations. All features, both in the global state and in individual observations of agents, are normalized by their maximum values.

**Action Space** The discrete set of actions which agents are allowed to take consists of move[direction], attack[enemy id], stop and no-op. Dead agents can only take no-op action while live agents cannot. Agents can only move with a fixed movement amount 2 in four directions: north, south, east, or west. To ensure decentralization of the task, agents are restricted to use the attack[enemy id] action only towards enemies in their shooting range. This additionally constrains the ability of the units to use the built-in attack-move macro-actions on the enemies that are far away. The shooting range is set to be6 for all agents. Having a larger sight range than a shooting range allows agents to make use of the move commands before starting to fire. The unit behavior of automatically responding to enemy fire without being explicitly ordered is also disabled.

**Rewards** At each time step, the agents receive a joint reward equal to the total damage dealt on the enemy units. In addition, agents receive a bonus of 10 points after killing each opponent, and 200 points after killing all opponents for winning the battle. The rewards are scaled so that the maximum cumulative reward achievable in each scenario is around 20.

**Environmental Settings of RMIX** The difficulty level of the built-in game AI we use in our experiments is level 7 (very difficult) by default as many previous works did (Rashid et al., 2018; Mahajan et al., 2019; Yang et al., 2020). The scenarios used in Section 4 are shown in Table 1. We present the table of all scenarios in SMAC in Table 1 and the corresponding memory usage for training each scenario in Table 2. The *Ally Units* are agents trained by MARL methods and *Enemy Units* are built-in game bots. For example, 5m_vs_6m indicates that the number of MARL agent is 5 while the number of the opponent is 6. The agent (unit) type is *marine*[7]. This asymmetric setting is hard for MARL methods.

---

[7]A type of unit (agent) in StarCraft II. Readers can refer to `https://liquipedia.net/starcraft2/Marine_(Legacy_of_the_Void)` for more information

Table 1: SMAC Environments

| Name | Ally Units | Enemy Units | Type |
|------|-----------|-------------|------|
| 3m | 3 Marines | 3 Marines | homogeneous & symmetric |
| 8m | 8 Marines | 8 Marines | homogeneous & symmetric |
| 25m | 25 Marines | 25 Marines | homogeneous & symmetric |
| 3s5z | 3 Stalkers & 5 Zealots | 3 Stalkers & 5 Zealots | heterogeneous & symmetric |
| MMM | 1 Medivac, 2 Marauders & 7 Marines | 1 Medivac, 2 Marauders & 7 Marines | heterogeneous & symmetric |
| 5m_vs_6m | 5 Marines | 6 Marines | homogeneous & asymmetric |
| 8m_vs_9m | 8 Marines | 9 Marines | homogeneous & asymmetric |
| 10m_vs_11m | 10 Marines | 11 Marines | homogeneous & asymmetric |
| 27m_vs_30m | 27 Marines | 30 Marines | homogeneous & asymmetric |
| 3s5z_vs_3s6z | 3 Stalkers & 5 Zealots | 3 Stalkers & 6 Zealots | heterogeneous & asymmetric |
| MMM2 | 1 Medivac, 2 Marauders & 7 Marines | 1 Medivac, 3 Marauders & 8 Marines | heterogeneous & asymmetric |
| 1c3s5z | 1 Colossi & 3 Stalkers & 5 Zealots | 1 Colossi & 3 Stalkers & 5 Zealots | heterogeneous & symmetric |
| 2m_vs_1z | 2 Marines | 1 Zealot | micro-trick: alternating fire |
| 3s_vs_5z | 3 Stalkers | 5 Zealots | micro-trick: kiting |
| 6h_vs_8z | 6 Hydralisks | 8 Zealots | micro-trick: focus fire |
| corridor | 6 Zealots | 24 Zerglings | micro-trick: wall off |
| bane_vs_bane | 20 Zerglings & 4 Banelings | 20 Zerglings & 4 Banelings | micro-trick: positioning |

## F  ADDITIONAL TRAINING DETAILS

The baselines are list in table 3 as depicted below. To make a fair comparison, we use `episode` (single-process environment for training, compared with `parallel`) runner defined in PyMARL to run all methods. The evaluation interval is $10,000$ for all methods. We use uniform probability to estimate $Z_i(\cdot, \cdot)$ for each agent. We use the other hyper parameters used for training in the original papers of all baselines. The metrics are calculated with a moving window size of 15. Experiments are carried out on NVIDIA Tesla V100 GPU 16G. We also provide memory usage of baselines (given the current size of the replay buffer) for training each scenario of SCII domain in SMAC.

We use the same neural network architecture of agent used by QMIX (Rashid et al., 2018). The trajectory embedding network $\phi_i$ is similar to the network of the agent.

Table 2: Memory usage (given the current size of the replay buffer) for the training of each method (exclude COMA, which is an on-policy method without using replay buffer) on scenarios of SCII domain in SMAC.

| Scenario Name | Memory Usage (GB) |
|---|---|
| 3m | 2.7 |
| 2m_vs_1z | 2.8 |
| 5m_vs_6m | 3 |
| 3s_vs_5z | 4 |
| 6h_vs_8z | 4.6 |
| 8m | 4.8 |
| 8m_vs_9m | 4.9 |
| 3s5z | 6.4 |
| 10m_vs_11m | 7.1 |
| 3s5z_vs_3s6z | 7.5 |
| 1c3s5z | 8.6 |
| MMM | 8.7 |
| MMM2 | 10.8 |
| corridor | 14.4 |
| 25m | 27 |
| 27m_vs_30m | 39.5 |
| bane_vs_bane | 41 |

Table 3: Baseline algorithms

| Algorithms | Brief Description |
|---|---|
| IQL (Tampuu et al., 2017) | Independent Q-learning |
| VDN (Sunehag et al., 2017) | Value decomposition network |
| COMA (Foerster et al., 2017a) | Counterfactual Actor-critic |
| QMIX (Rashid et al., 2018) | Monotonicity Value decomposition |
| QTRAN (Son et al., 2019) | Value decomposition with linear affine transform |
| MAVEN (Mahajan et al., 2019) | MARL with variational method for exploration |
| Qatten (Yang et al., 2020) | Multi-head attention for decomposing the global Q values |

## G  ADDITIONAL EXPERIMENTS ON SMAC

### G.1  ABLATIONS

We use the same hyper-parameters in ablation study unless otherwise specified.

#### G.1.1  STATIC RISK LEVEL

We present more results of RMIX and RMIX ($\alpha = 1$) in Figure 13, 14 and 15. In **vary hard** 3s5z_vs_3s6z, 6h_vs_8z games, RMIX and RMIX ($\alpha = 1$) outperforms baselines. Surprisingly, RMIX is even slightly better in 6h_vs_8z where micro-trick (focus fire) is learned. In conclusion, RMIX is also capable of learning hard micro-trick tasks in StarCraft II. In *asymmetric* scenario 27m_vs_30m, RMIX shows leading performance as well.

We also conduct an ablation study with static risk level of $\alpha = 0.1$ and $\alpha = 0.3$ in RMIX-static. Obviously, as shown in Figure 16, with static risk level, RMIX-static shows steadily progress over time, but its performance is lower than RMIX in 6h_vs_8z and 5m_vs_6m. On *asymmetric* scenario 5m_vs_6m, the converged performance of RMIX-static is 0.6, which is lower than RMIX's (0.8).

In some scenarios, where risk is not the primal concern for policy optimization, for example corridor scenario, where agents should learn to explore to win the game. RMIX shows less sample efficient to learn the optimal policy compared with its risk neutral variant, RMIX ($\alpha = 1$) in corridor as shown in Figure 16(b). RMIX learns slower than RMIX ($\alpha = 1$) because it relies on the dynamic risk predictor while the predictor is trained together with the agent, it will take more samples in these environments. Interestingly, RMIX shows very good performance over other baselines.

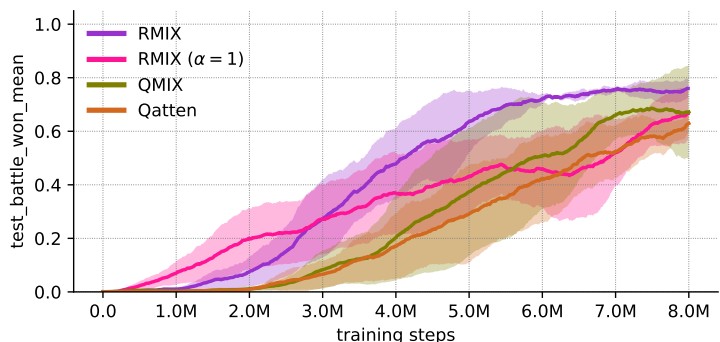

Figure 13: `test_battle_won_mean` of RMIX, RMIX ($\alpha = 1$) and QMIX on 3s5z_vs_3s6z (*heterogeneous* and *asymmetric* scenario, very **hard** game).

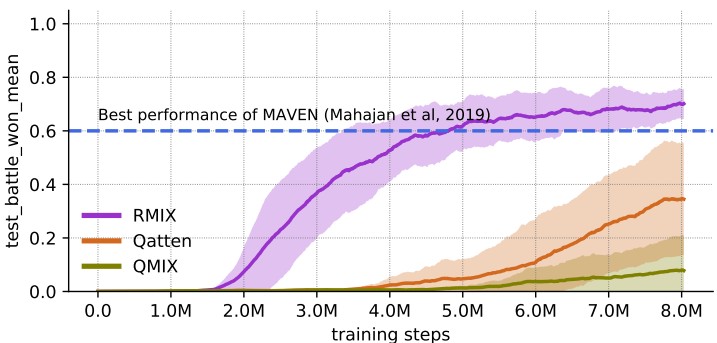

Figure 14: `test_battle_won_mean` of RMIX, RMIX ($\alpha = 1$) and QMIX on 6h_vs_8z (*homogeneous* and *asymmetric* scenario, **very hard** game)

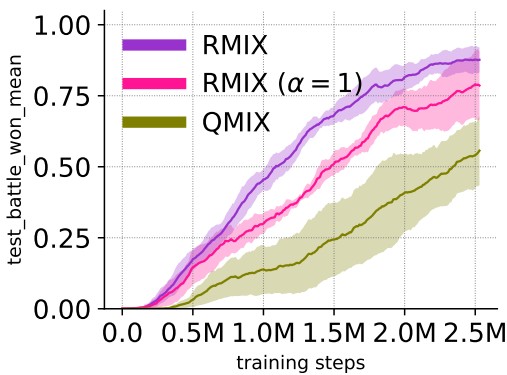

Figure 15: `test_battle_won_mean` of RMIX, RMIX ($\alpha = 1$) and QMIX on 27m_vs_30m (homogeneous and asymmetric scenario, **very hard** game).

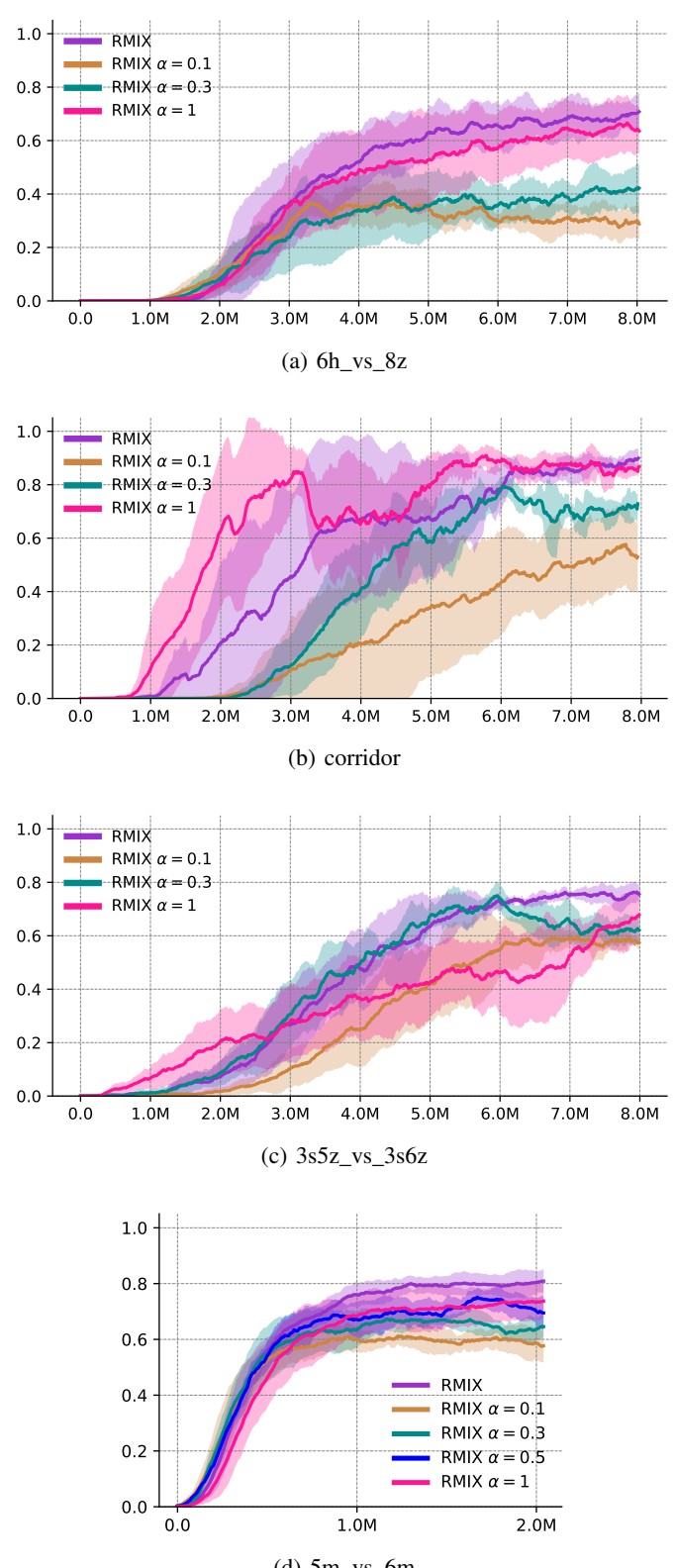

(a) 6h_vs_8z

(b) corridor

(c) 3s5z_vs_3s6z

(d) 5m_vs_6m

Figure 16: `test_battle_won_mean` of RMIX-static.

Table 4: Hyper-parameters

| hyper-parameter | Value |
|---|---|
| Batch-size | 32 |
| Replay memory size | 5,000 |
| Optimizer | Adam |
| Learning rate (lr) | 5e-4 |
| Critic lr | 5e-4 |
| RMSProp alpha | 0.99 |
| RMSProp epsilon | 0.00001 |
| Gradient norm clip | 10 |
| Action-selector | $\epsilon$-greedy |
| $\epsilon$-start | 1.0 |
| $\epsilon$-finish | 0.05 |
| $\epsilon$-anneal-time | 50,000 steps |
| Target-update-interval | 200 |
| Evaluation interval | 10,000 |
| Number of atoms (m) | 55 |
| K | 10 |
| Runner | `episode` |
| Training steps | 1, 2, 3, 1.5, 2.5 and 8 millions |
| Discount factor ($\gamma$) | 0.99 |
| RNN hidden dim | 64 |

## G.2 ADDITIONAL RESULTS ANALYSIS

We provide additional results analysis of RMIX on corridor in Figure 17. There are 6 RMIX agents in corridor. For brevity of visualization, we use the data of 3 agents (agent 0, 1 and 3) to analyse the results and to demonstrate RMIX agents have learned to address time-consistency issue.

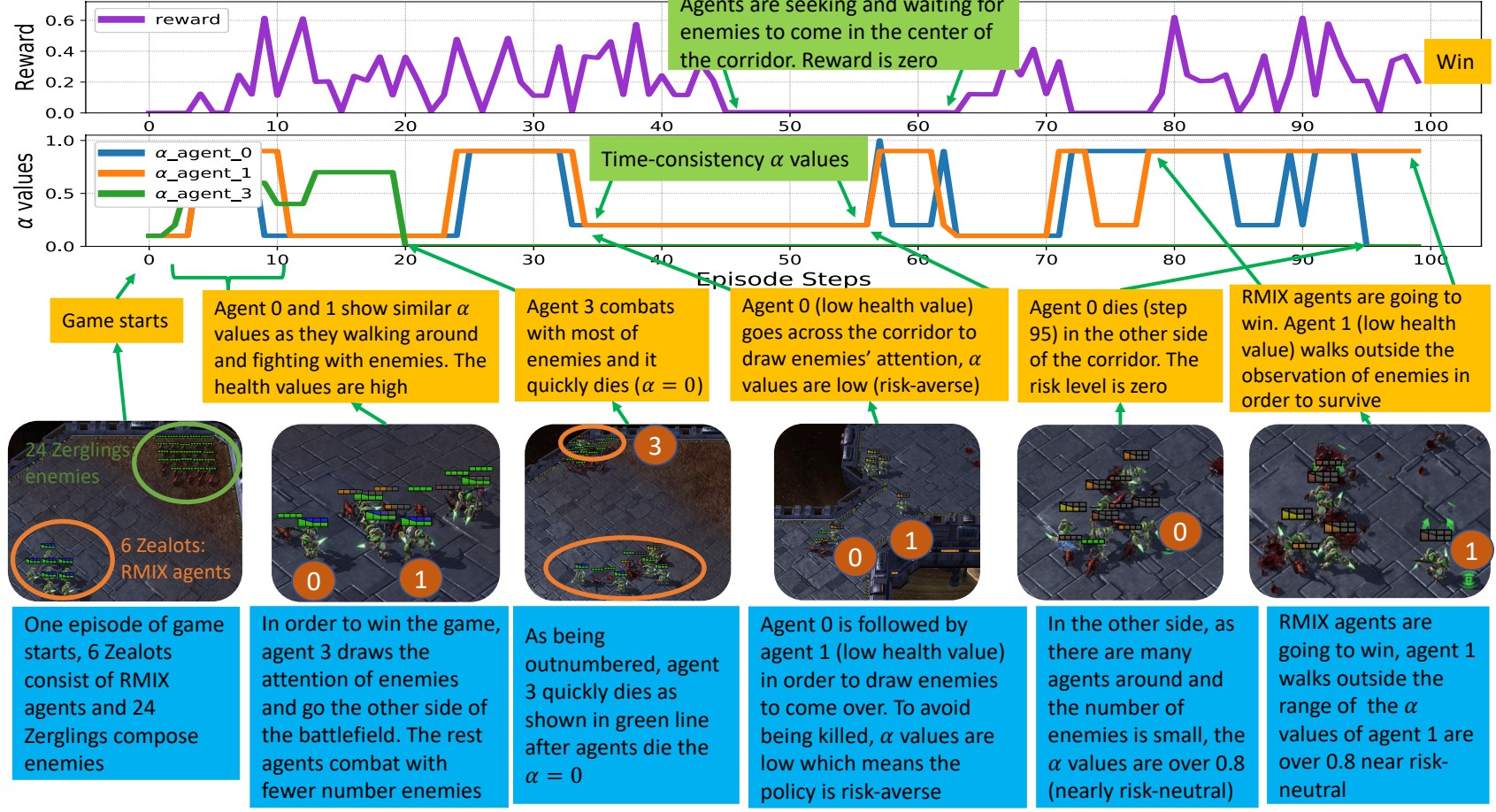

Figure 17: RMIX results analysis on corridor. We use trained model of RMIX and run the model to collect one episode data including game replay, states, actions, rewards and $\alpha$ values (risk level). We show rewards of one episode and the corresponding $\alpha$ value each agent predicts per time step in row one and row two. We provide description and analyses on how agents learn time-consistency $\alpha$ values for the rest rows. Pictures are screenshots from the game replay. Readers can watch the game replay via this anonymous link: https://youtu.be/J-PG0loCDGk. Interestingly, it also shows emergent cooperation strategies between agents at different time step during the episode, which demonstrate the superiority of RMIX.

## G.3 ADDITIONAL RESULTS

We conduct experiments of RMIX, QMIX, MAVEN, Qatten, VDN, COMA and IQL on 17 SMAC scenarios. We show results of `test_battle_won_mean` and `test_return_mean` of afore-mentioned methods in Figure 19 and 20, respectively. RMIX shows leading performance on most of scenarios, ranging from *symmetric homogeneous* scenarios to *asymmetric heterogeneous* scenarios. Surprisingly, RMIX also shows superior performance on scenarios where *micro-trick* should be learned to win the game.

In addition, we compare RMIX with QR-MIX (Hu et al., 2020). Unlike QMIX, QR-MIX decomposes the estimated joint return distribution into individual Q values. We implement QR-MIX with PyMARL by using hyper-parameters in QR-MIX paper and train it on 3m (easy), 1c3s5z (easy), 5m_vs_6m (hard), 8m_vs_9m (hard), 10_vs_11m (hard), 27m_vs_30m (very hard), MMM2 (very hard), 3s5z_vs_3s6z (very hard), corridor (very hard) and 6h_vs_8z (very hard) with 3 random seeds for each scenario. Results are shown in Figure 18.

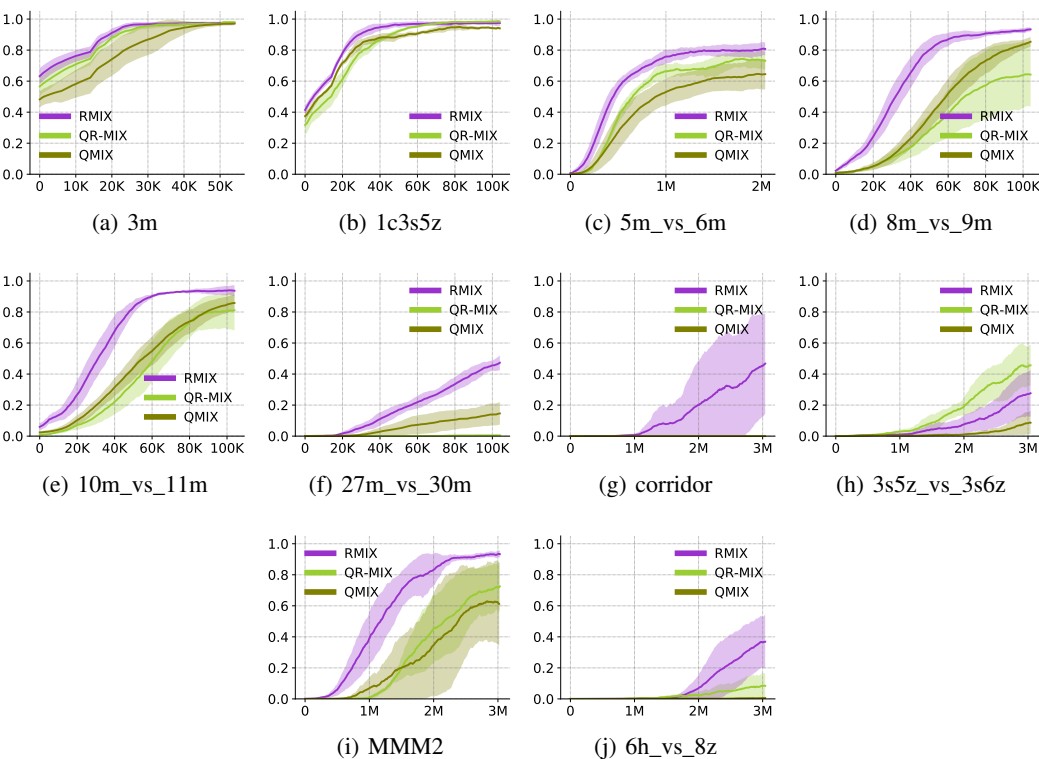

Figure 18: `test_battle_won_mean` of RMIX vs QR-MIX and QMIX.

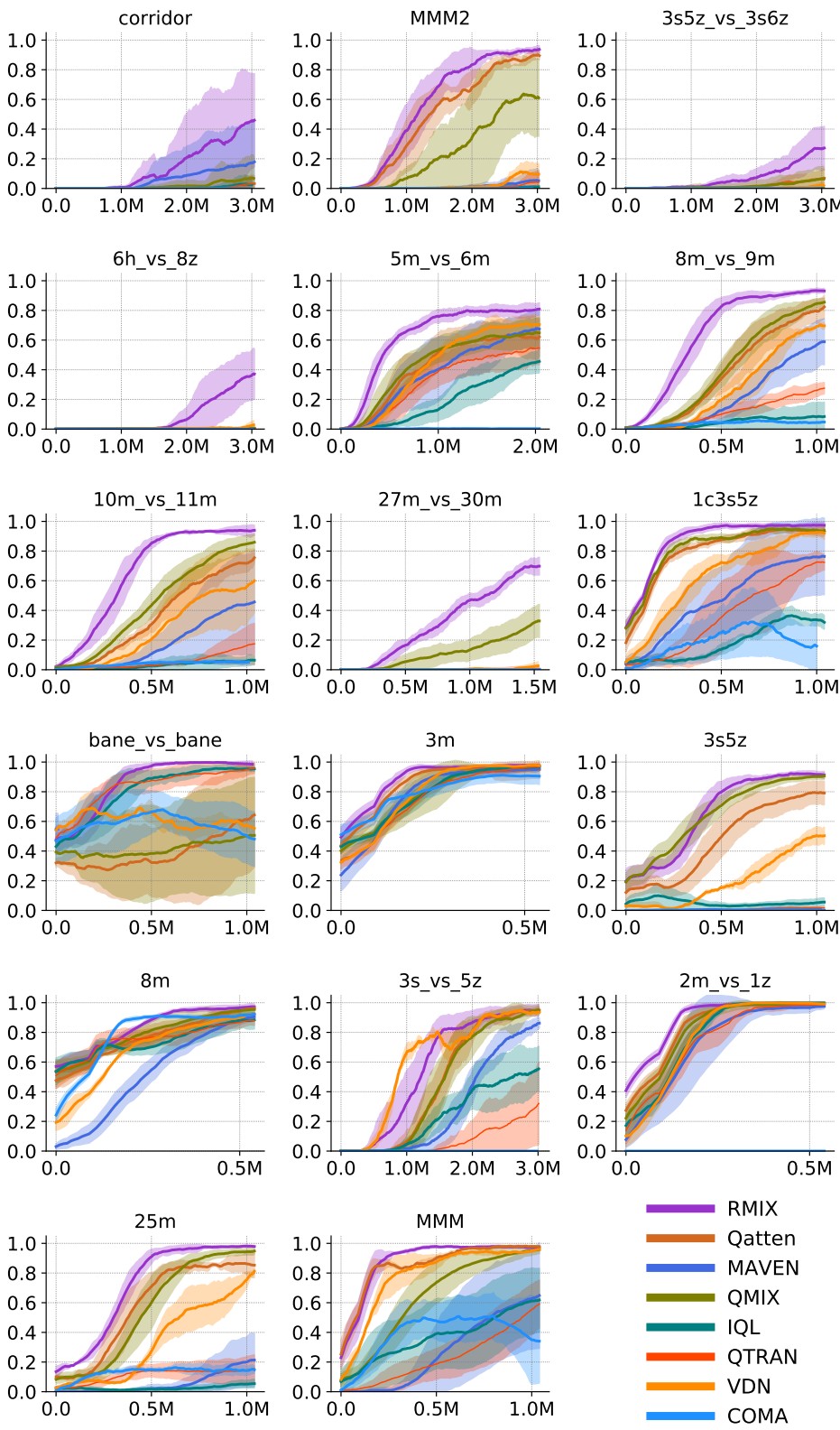

Figure 19: `test_battle_won_mean` of RMIX, MAVEN, QMIX, Qatten, IQL, QTRAN, VDN and COMA on 17 SMAC StarCraft II scenarios.

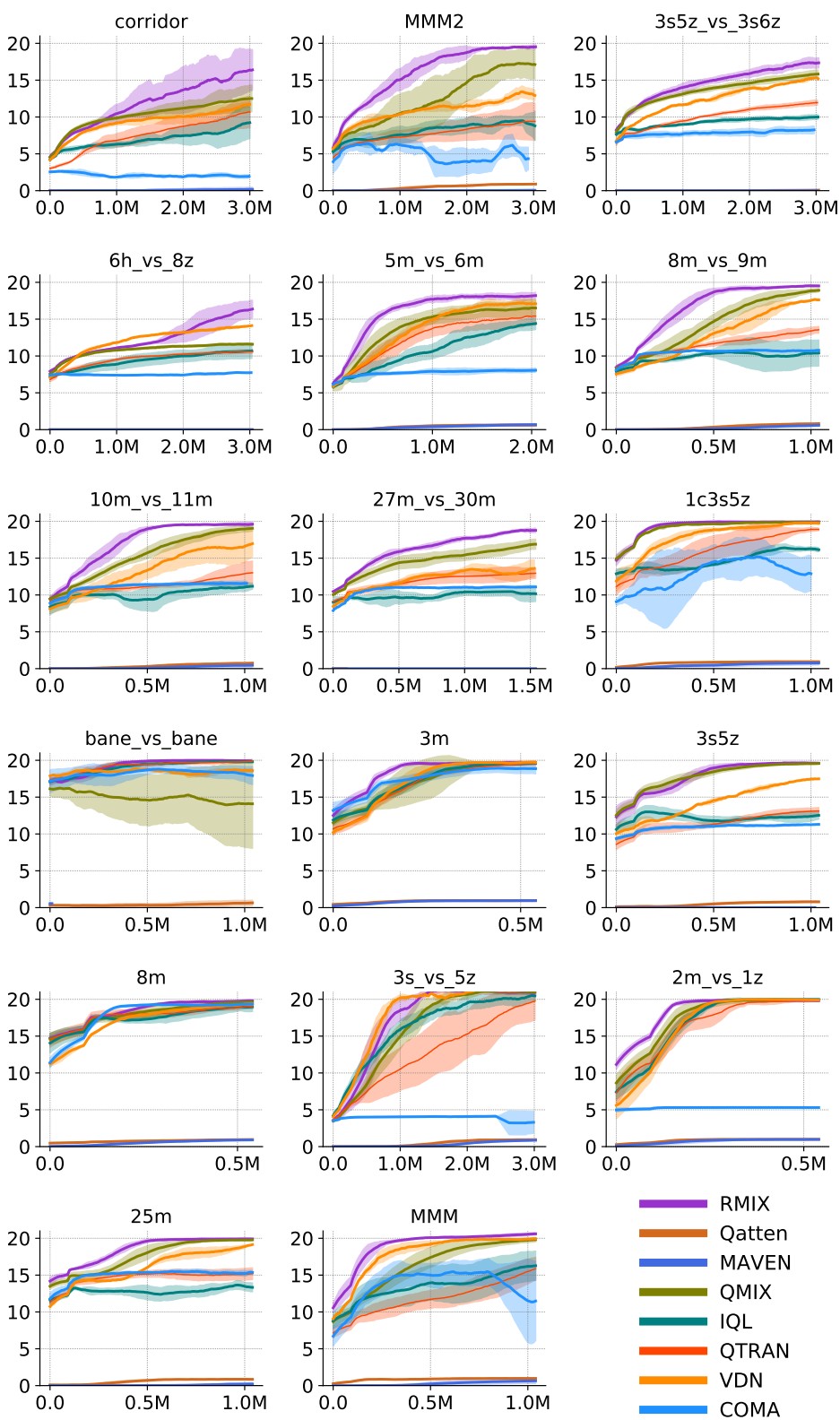

Figure 20: `test_return_mean` of RMIX, MAVEN, QMIX, Qatten, IQL, QTRAN, VDN, COMA on 17 SMAC StarCraft II scenarios.

