# OpenReview forum: "RMIX: Risk-Sensitive Multi-Agent Reinforcement Learning"
_ICLR.cc/2021/Conference — Reject_

### Official Review · AnonReviewer3 · 2020-10-22
**Reviews**

**Rating:** 6
**Confidence:** 2

**Review:**

The authors propose RMIX to deal with the randomness of rewards and the uncertainty in environments. RMIX learns the individual value distributions of each agent and uses a predictor to calculate the dynamic risk level. Given the individual value distribution and the risk level, a CVaR operator outputs the C value for execution. For training, the $C$ values are mixed as $C^{tot}$ and updated by TD error end-to-end. RMIX outperforms a series of value decomposition baselines on many challenging StarCraft II tasks. The paper is very clear and well-structured. Expanding value decomposition methods to the risk-sensitive field is a novel idea, and it shows competitive performance in empirical studies.

However, my main concern is the definition of the risk level $\alpha$. More in-depth analysis is expected to interpret why the discrepancy between the embedding of current individual return distributions and the embedding of historical return distributions could reflect the risk level. Since the embedding parameters $femb$ and $\phi$ are trained end-to-end by TD-error, the real meaning of $\alpha$ is unknowable. Eq. 3 is a little confusing, and I get the left side of Eq. 3 should be $p_{\alpha_i^k}$, not $\alpha_i^k$. It is hard to understand how to get the final $\alpha_i$ from the K-range probability. Figure 12 helps a lot in this understanding, and I expect it to be more detailed and put on the main pages.

RMIX is built on QMIX. However, the individual value function in QMIX does not estimate a real expected return, and the value has no meaning. Is the theoretical analysis of the risk still valid in QMIX？

RMIX is proposed for the randomness of rewards and the uncertainty in environments. However, I think they are not usually observed in the environment StarCraft II, since the policies of the enemy are fixed. Increasing the randomness and uncertainty could verify the advantages of RMIX.

----------Update after author response----------

I thank the authors for the detailed response. Most of my concerns have been addressed, and I decide to keep my score.

---

> ### Author Response · Authors · 2020-11-20
> **Responses to AnonReviewer3 (part 1)**
>
> Dear reviewer, we thank you for providing very valuable suggestions. We will explain your concerns and answer your questions point by point.
>
> ---
>
> **Q1: the definition of the risk level $\alpha$. More in-depth analysis is expected to interpret why the discrepancy between the embedding of current individual return distributions and the embedding of historical return distributions could reflect the risk level.**
>
> **A1**:
> (1) $\alpha$ (a hyper-parameter) is a risk level ($\alpha \in (0, 1]$) in CVaR and it can be static or dynamic.
>
> (2) The differences between the current individual return distributions and the local historical distributions are that the local historical distributions do not have the current individual return distributions as embedding and features are learned by GRU with trajectories as inputs.
>
> (3) Intuitively, the current local return distribution maybe different from the local distribution of the last step. If the local return distribution changes a little, the risk level would not change a lot. Therefore, we can model this change and measure the discrepancy.
>
> (4) As shown in Figure 3, with two GRUs modelling the current local return distribution and the historical local return distribution (here, we use the local return distribution of the last step as the historical local return distribution as the GRU has memorised the features of historical return distribution). Therefore, by utilizing the features, we can use the embedding of current individual return distributions and the embedding of historical return distributions to predict the risk level.
>
> ---
>
> **Q2: Since the embedding parameters $f_{emb}$ and $\phi$ are trained end-to-end by TD-error, the real meaning of  $\alpha$ is unknowable.**
>
> **A2**: The $\alpha$ is a parameter in CVaR. It presents the risk level in CVaR. The learned $\alpha$ is used as risk level for CVaR calculation in decentralized training. As we use CVaR on return distributions, it corresponds to risk-neutrality (expectation) and indicates the improving degree of risk-aversion when the risk level $\alpha_i=1$ and $\alpha_i \rightarrow 0$, respectively.
>
> ---
>
> **Q3: Eq. 3 is a bit confusing, and I get the left side of Eq. 3 should be $p_{\alpha_i^k}$, not $\alpha_i^k$. It is hard to understand how to get the final $\alpha_i$ from the K-range probability. Figure 12 helps a lot in this understanding, and I expect it to be more detailed and put on the main pages.**
>
> **A3**: We agree it is confusing. The left side should be probability of $p_{\alpha_i^k}$. As we can use Eq. 3 to calculate the probability of each risk level. can get a probability vector containing K probability values, and we can get the  $k \in [1, …, K]$ with the maximal probability via argmax and normalize $k$ into $(0, 1]$, thus  $\alpha_i$ is calculated.. We can use this value to mask out parts of local return distribution out of the risk level to get the final CVaR value. We updated this part in the main text.
>
> ---
>
> **Q4: RMIX is built on QMIX. However, the individual value function in QMIX does not estimate a real expected return, and the value has no meaning. Is the theoretical analysis of the risk still valid in QMIX?**
>
> **A4**:
> The theoretical analysis of the risk is still valid.
>
> (1) In Dec-POMDP-based problems, the reward at each time step is a global reward which is shared by each agent and each agent has no access to its real contribution to the global reward. VDN, QMIX and many other MARL algorithms [1,2,3,4] were proposed to address this credit assignment issue in Dec-POMDP-based problems.
>
> (2) Concretely, the global values are factorized (sum of individual values in VDN and monotonicity in QMIX) to each individual value and agents can use these individual values to get execution policies for decentralized execution. During execution, at each time step, each agent takes its local observation and previous time step actions as inputs to the agent network to get the local estimated values which are further used as policies.
>
> (3) Therefore, the global value factorization estimates the local value. In RMIX, we also use the monotonic network for centralized training and the risk operator works over the local estimated return distribution. IGM and monotonic value network can be applied to many Dec-POMDP-based MARL algorithm and the theoretical analysis of the risk still valid.

---

> > ### Author Response · Authors · 2020-11-20
> > **Responses to AnonReviewer3 (part 2)**
> >
> > **Q5: However, I think they are not usually observed in the environment StarCraft II, since the policies of the enemy are fixed. Increasing the randomness and uncertainty could verify the advantages of RMIX.**
> >
> > **A5**:
> > Your observation truly reflects the mechanism of StarCraft II (SCII).  The built-in bots (enemies) in SCII use heuristic methods, so policies of the enemy are not always fixed as the policies of enemies adapt to the actions of MARL agents. Moreover, as each agent acting independently with local partial observation as inputs, it also increases the randomness and uncertainty, thus leading to non-stationarity which make training MARL methods challenging.
> >
> > ---
> >
> > **References**:
> >
> > [1] Son, Kyunghwan, et al. "QTRAN: Learning to Factorize with Transformation for Cooperative Multi-Agent Reinforcement Learning." International Conference on Machine Learning. 2019.
> >
> > [2] Rashid, Tabish, et al. "QMIX: Monotonic Value Function Factorisation for Deep Multi-Agent Reinforcement Learning." International Conference on Machine Learning. 2018.
> >
> > [3] Yang, Yaodong, et al. "Q-value Path Decomposition for Deep Multiagent Reinforcement Learning." International Conference on Machine Learning. 2020.
> >
> > [4] Sunehag, Peter, et al. "Value-decomposition networks for cooperative multi-agent learning." arXiv preprint arXiv:1706.05296 (2017).

---

### Official Review · AnonReviewer4 · 2020-10-27
**Important issue for MARL, yet lack of clarity**

**Rating:** 6
**Confidence:** 3

**Review:**

Overall
The paper considers the cooperative multiagent MARL setting where agents have partial observability and share a team reward. Instead of optimizing the expected cumulative reward, the paper considers optimizing the CVaR measure. The proposed method relies on CTDE and features a dynamic risk-level predictor. Although CVaR optimization is an important problem for MARL and the experimental results seem to be convincing, the formulation of the problem and the description of the proposed method are not written with enough clarity. Specifically, I have the following comments/questions.


Comments/Questions

1. For CVaR optimization, is the risk level (i.e. alpha) given as part of the problem itself? If it is given, what’s the intuition behind the “dynamic risk level predictor” that handles “the temporal nature of stochastic outcomes”. What is “the temporal nature”?

2. It’s known that, for a discrete random variable, its pdf is composed by Dirac functions weighted by the pmf. Is Definition 1 repeating this?

3. I have difficulty understanding Equations (1) and (2). In (1), what’s the meaning of delta(tau_i, u_i)? Isn’t that the Dirac function takes a number as its argument? Why can the Dirac function take a trajectory as its argument? For (2), why does the estimate not depend on P_j that appears in (1)?

4. In Theorem 1, what is the definition of IGM, and the definition of C^tot? If the reward is shared, why would the local CVaR be different from the global CVaR?


Although the paper considers an important optimization objective for MARL, I don’t think the presentation is ready for publication yet.

---

> ### Author Response · Authors · 2020-11-20
> **Responses to AnonReviewer4**
>
> Dear reviewer, we thank you for your very helpful comments. We appreciate you recognise the significance of the risk-sensitive problem in MARL. These are some parts we did not clarify in our paper. To enhance the clarity, we have improved the paper. We will explain your concerns and answer your questions point by point.
>
> ---
>
> **Q1: is the risk level (i.e. alpha) given as part of the problem itself? If it is given, what’s the intuition behind the “dynamic risk level predictor” that handles “the temporal nature of stochastic outcomes”. What is “the temporal nature”?**
>
> **A1**:
>
> (1) The risk level is not a problem. It is a parameter in CVaR and it can be static (predefined) or dynamic.
>
> (2) The motivation behind the “dynamic risk level predictor” is that there are some consecutive states where the risk level should be fixed in high range while for other consecutive states, the risk level should stay in other low range. Assume a team of marines pass a long corridor where the number of enemies is larger than that of marines. Under such circumstances, the risk level should be low (As we use CVaR on return distributions, it corresponds to risk-neutrality (expectation, $\alpha_i=1$) and indicates the improving degree of risk-aversion ($\alpha_i \rightarrow 0$)) as marines should focus on the worst cases (return is low), that is the whole army is (completely) wiped out. If the marines successfully passed the corridor and the environment is much safer then marines can explore and the risk level is low. This is the time-consistency issue [1,2] and is an issue in risk-sensitive RL. We also provide an analysis in Figure 17, Appendix G.2 in the revised paper.
>
> (3) Temporal nature means the characteristic of stochastic rewards related to time.
>
> ---
>
>
> **Q2: It’s known that, for a discrete random variable, its pdf is composed by Dirac functions weighted by the pmf. Is Definition 1 repeating this?**
>
> **A2**: Yes. Definition 1 defines the generalized return PDF of each agent given the trajectories of each agent and Dirac functions are parameterized by neural networks.
>
> ---
>
> **Q3: For (2), why does the estimate not depend on P_j that appears in (1)?**
>
> **A3**: We thank the reviewer for finding it. Yes, the calculation of CVaR depends on the $P_j$. We have update the paper on this part to improve clarity.
>
> ---
>
> **Q4: In (1), what’s the meaning of delta(tau_i, u_i)? Isn’t that the Dirac function takes a number as its argument? Why can the Dirac function take a trajectory as its argument? For (2), why does the estimate not depend on P_j that appears in (1)?**
>
> **A4**: Yes, Dirac function takes a number as its argument. The reason is that we want to generalize it with agent’s experiences as the high dimension input as widely used in MARL methods [3,4]. While some other papers may use observations [5] or the hidden states [6]. Parameterized by GRU, we use the hidden states (numbers) as inputs to output the Dirac outputs.
>
> ---
>
> **Q5: what is the definition of IGM, and the definition of $C^{tot}$?**
>
> **A5**:  QTRAN [3] introduced IGM. IGM says the optimal joint actions across agents are equivalent to the collection of individual optimal actions of each agent. This principle is used for global value factorization. $C^{tot}$ is the output from the monotonic network by taking the individual CVaR values as inputs.
>
> ---
>
> **Q6: If the reward is shared, why would the local CVaR be different from the global CVaR?**
>
> **A6**: The shared reward is a global reward and each agent has no access to its real contribution (local reward). Therefore, credit assignment is needed to estimate the contribution of each agent and training the policy of each agent. VDN [6] and QMIX [4] were proposed to address this problem. In RMIX, each agent also has no access to its real contribution to the global reward and instead gets the global reward. Possessing various kinds of abilities in each agent, the local CVaR values, which compose the global CVaR values via a monotonic network, are different from the global CVaR values.
>
> ---
>
> **References**:
>
> [1] Ruszczyński, Andrzej. "Risk-averse dynamic programming for Markov decision processes." Mathematical programming 125.2 (2010): 235-261.
>
> [2] Tamar, Aviv, et al. "Policy gradient for coherent risk measures." Advances in Neural Information Processing Systems. 2015.
>
> [3] Son, Kyunghwan, et al. "QTRAN: Learning to Factorize with Transformation for Cooperative Multi-Agent Reinforcement Learning." International Conference on Machine Learning. 2019.
>
> [4] Rashid, Tabish, et al. "QMIX: Monotonic Value Function Factorisation for Deep Multi-Agent Reinforcement Learning." International Conference on Machine Learning. 2018.
>
> [5] Yang, Yaodong, et al. "Q-value Path Decomposition for Deep Multiagent Reinforcement Learning." International Conference on Machine Learning. 2020.
>
> [6] Sunehag, Peter, et al. "Value-decomposition networks for cooperative multi-agent learning." arXiv preprint arXiv:1706.05296 (2017).

---

> > ### Comment · AnonReviewer4 · 2020-11-22
> > **Thanks for the response**
> >
> > I've increased my score.

---

> > > ### Author Response · Authors · 2020-11-23
> > > **Response to AnonReviewer4: thanks for raising the score**
> > >
> > > We thank AnonReviewer4 for the response and raising the score.

---

### Official Review · AnonReviewer1 · 2020-10-27
**Theoretical aspects could be stronger, but clear demonstration of the practical merits in ambitious experiments.**

**Rating:** 7
**Confidence:** 4

**Review:**


Strengths:

1) The paper is well-written, and versed in the pulse of related works on the topic. This makes it very easy to assess the points of differentiation of the formulation/focus from prior art. Especially the problem class of MARL with risk measures is salient.

2)  The authors clearly demonstrate the practical merits of the proposed techniques in ambitious experiments. This is a major upshot of the work: one may clearly see across challenging domains the performance gains achieved by RMIX.

3) Solving Dec-POMDPs is in general intractable, and one must often resort to particle filtering/belief representations of the state. Combining these technical challenges with risk sensitive utilities is nontrivial, and the authors have made a bold effort to obtain a tractable algorithm despite these issues. Importantly, their work has some conceptual justification.



Weaknesses:


1) While the conceptual setup and theoretical contributions are clear, the actual training mechanism is given very little explanation. There should at least be an iterative update scheme or pseudo-code presenting the key algorithm of this work. This would serve to make it easier to distinguish what are the unique attributes to the algorithm put forth in this work outside of a contextual discussion and at a more granular algorithmic level. By reading the paper it is very difficult to understand what information agents must exchange during training.

2) What is the purpose of the generalized return PDF? How does the information required for its estimation get processed algorithmically, and can each agent estimate its component of it with local information only? This is not easy to discern.

3) Theorem 1 must somehow assume that each agent's marginal utility is concave and that the joint utility is jointly concave in agents' local policies, but a discussion of this seems missing. Without concavity, then one can only ensure that agents' policies in the decentralized setting converge to stationarity, and that these stationary points belong to the set of extrema of the global utility. It may be the case that the fact that the risk-sensitive Bellman operator being a contraction is enough to mitigate these subtleties and ensure convergence to a global extrema, but this is not discussed.

4) Just defining the TD loss in eqn. (4) is not enough because the presence of a risk measure means that a vanilla TD step in terms of approximating an expected value does not hold. How does the `````blocking to avoid
changing the weights of the agents’ network from the dynamic risk level predictor" change the TD population objective? Under normal circumstances, this would be the Bellman optimality operator -- how does C^{tot} mitigate this issue, and what relationship does this hold to the notions of risk-sensitive Bellman equations in

Andrzej Ruszczy´nski. Risk-averse dynamic programming for markov decision processes. Mathematical
Programming, 125(2):235–261, 2010.

See also:

Kose, U., & Ruszczynski, A. (2020). Risk-Averse Learning by Temporal Difference Methods. arXiv preprint arXiv:2003.00780.

5) In general, a discussion of the technical innovations required to establish the theorems is absent. Are these theorems inherited from the algorithms appearing in earlier work? What is new?

Minor Comments:

1) References missing regarding risk-sensitive RL:

Zhang, Junyu, Amrit Singh Bedi, Mengdi Wang, and Alec Koppel. "Cautious Reinforcement Learning via Distributional Risk in the Dual Domain." arXiv preprint arXiv:2002.12475 (2020).

2) The link between equation (2) and estimating a conditional expectation should be made more explicit.

3) The meaning and interpretation of IGM is ambiguous. More effort needs to be expended to properly explain it. The idea seems to be that individual agents' optimal policy is equivalent to an oracle/meta-agent that aggregates information over the network. This notion is then key to interpreting the importance of Theorem 1.

---

> ### Comment · AnonReviewer1 · 2020-11-10
> **Open-minded if someone makes a technical claim that this paper should be rejected, but demanding comparison distributional RL is unfair.**
>
> Can anyone give a credible technical reason this paper should not be accepted? To my knowledge, considering a global risk measure over agents' rewards yields a non-node-separable problem. There was some ambiguity in how this was addressed, except that a degree of node separability is assumed. This, however, would invalidate the notion of a global CVaR over all agents' local rewards.
>
> I am looking for more comments of this type in order to consider reducing my score.

---

> ### Author Response · Authors · 2020-11-20
> **Responses to AnonReviewer1 (part 1)**
>
> Dear reviewer, we appreciate your high-quality reviews and constructive feedbacks. We have cited the new risk-sensitive papers in our paper. Due to the 8-page main text limit, there are some parts are not discussed in detail and we have clarified them in the paper. Our goal is to design a risk-sensitive MARL method to fill the gap as few works have been done and the risk-sensitive problem is also a significant problem in real world applications. We will address your concerns and answer your questions point by point.
>
> ---
>
> **Q1: the actual training mechanism is given very little explanation**
>
> **A1**: The training is centralized training. The target of RMIX is to update the weights in agent network, risk operator and monotonic network.
>
> (i) First, a mini-batch of experience (state, observations, actions, rewards, next states) collected by each agent at each step is sampled from the replay buffer.
>
> (ii) Then we calculate the individual $C^{t}_{i}$  of each agent by the equation in section 3.1.
>
> (iii) Third, we input the $C^{t}_{i}$ into the monotonic network and get the $C^{tot}$. By following this step, we can get the target value and the predicted value in equation 7 and thus we get the TD loss.
>
> (iv) Finally, we can use gradient descent methods to minimize the TD loss. We also have added pseudo code Algorithm 1 in Appendix in the revised paper.
>
> ---
>
> **Q2: What is the purpose of the generalized return PDF?**
>
> **A2**: We define the generalized return PDF for the local return distribution modelled by many Dirac functions which are further used to estimate the CVaR values.
>
>
> ---
>
> **Q3: Theorem 1 must somehow assume that each agent's marginal utility is concave and that the joint utility is jointly concave in agents' local policies, but a discussion of this seems missing.**
>
> **A3**: We will put the discussion in the final version.
> (1) Theorem 1 is defined for decentralized execution in Centralized Training and Decentralized Execution (CTDE). In decentralized execution, each agent takes actions independently. As a consequence, there should be some rules designed for decentralized execution.
>
> (2) CVaR is a widely used risk measure and it enjoys many properties such as convexity (it can be converted into the concave form). Under the monotonic global CVaR value factorization, Theorem 1 holds and it can be used for decentralized execution.
>
> (3) Your comment on the assumption of the concavity of each agent’s marginal utility is insightful. It seems that there are few works discussing whether such concavity assumption exists in QMIX, QTRAN and other methods. If such assumption truly exists or there are some methods that can help to maintain this assumption, the representational capacity will be increased.
>
> ---
>
> **Q4: Just defining the TD loss in eqn. (4) is not enough because the presence of a risk measure means that a vanilla TD step in terms of approximating an expected value does not hold.**
>
> **A4**:
> The TD loss defined in Eqn. (4) is the centralized training loss function for updating the weights of the network of agents, risk operator and monotonic network. With risk measure, the CVaR values can be seen as the expectation value of local return values of the chosen region of the local return distribution given the current risk level. With such expectation values, the optimization of TD loss is minimized with expectation value from the chosen pessimistic values in the local return distribution.
>
> ---
>
> **Q5: How does the "blocking to avoid changing the weights of the agents’ network from the dynamic risk level predictor" change the TD population objective?**
>
> **A5**:
> In TD learning, both weights of risk level predictor and agent network should be updated. The motivation of this blocking gradient design (as shown in Figure 3) is to avoid gradients from risk level predictor to change the weights in the agent network because the agent network does not predict risk level and it estimates the local return distribution. This blocking gradient method is widely used in deep learning and deep RL, for example blocking gradients from the target network in DQN and its variants.

---

> > ### Author Response · Authors · 2020-11-20
> > **Responses to AnonReviewer1 (part 2)**
> >
> > **Q6: Under normal circumstances, this would be the Bellman optimality operator -- how does C^{tot} mitigate this issue, and what relationship does this hold to the notions of risk-sensitive Bellman equations in Andrzej Ruszczyński,. Risk-averse dynamic programming for markov decision processes. Mathematical Programming, 125(2):235–261, 2010.**
> >
> > **A6**:
> >
> > (1) $C^{tot}$ is the output from the monotonic network by taking the individual CVaR values as inputs. It is used for TD learning as each has no access to its contribution.
> >
> > (2) Andrzej Ruszczyński’s paper proposed the concept of a Markov risk measure and derived methods to solve the formulated risk-averse control problems. It seems that the risk-sensitive Bellman equation is mainly introduced in [1, 2] ([1, 2] are built on many previous works)
> >
> > (3) Our ideas were inspired by [1, 2] which introduce risk-sensitive Bellman equation and risk-sensitive Bellman for policy gradient. However, these works are on single-agent RL with tabular cases and CVaR is directly used to update the value function in order to minimize the cost. Simply applying them to multi-agent scenarios is a non-trivial task. We focus on Dec-POMDP-based MARL problem to maximize the global reward. Concretely, the global CVaR is estimated by taking all the individual CVaR values into the monotonic network and weights are updated in the TD loss. To make it work with deep learning with high dimensional inputs from complex multi-agent cooperation tasks, we also use GRU to model the dynamic risk measure which is not used in previous risk-sensitive RL works.
> >
> > ---
> >
> > **Q7: a discussion of the technical innovations required to establish the theorems is absent. Are these theorems inherited from the algorithms appearing in earlier work? What is new?**
> >
> > **A7**:
> >
> > (1) Theorem 1 is built on Definition 1 in [1] which introduces Individual-Global-MAX (IGM).
> >
> > (2) In our paper, with CVaR values as our policies, we have proved that the IGM property holds for CTDE under the monotonic factorization network.
> >
> > ---
> >
> > **Q8: The link between equation (2) and estimating a conditional expectation should be made more explicit.**
> >
> > **A8**:
> > We updated this part in the revised versin. Equation (2) is a close-form calculation of CVaR which is condition on values that are lower than the value at risk in Eqn (2).
> >
> > ---
> >
> > **Q9: The meaning and interpretation of IGM is ambiguous. More effort needs to be expended to properly explain it. The idea seems to be that individual agents' optimal policy is equivalent to an oracle/meta-agent that aggregates information over the network. This notion is then key to interpreting the importance of Theorem 1.**
> >
> > **A9**:
> > Individual-Global-MAX (IGM) says the optimal joint actions across agents are equivalent to the collection of individual optimal actions of each agent. In RMIX, the joint actions consist from optimal action of each agent via argmaxing the CVaR values. We proved it in the paper.
> >
> > ---
> >
> > **References**:
> >
> > [1] Chow, Yinlam, et al. "Risk-sensitive and robust decision-making: a cvar optimization approach." Advances in Neural Information Processing Systems. 2015.
> >
> > [2] Tamar, Aviv, et al. "Policy gradient for coherent risk measures." Advances in Neural Information Processing Systems. 2015.
> >
> > [3] Son, Kyunghwan, et al. "QTRAN: Learning to Factorize with Transformation for Cooperative Multi-Agent Reinforcement Learning." International Conference on Machine Learning. 2019.
> >
> > [4] Rashid, Tabish, et al. "QMIX: Monotonic Value Function Factorisation for Deep Multi-Agent Reinforcement Learning." International Conference on Machine Learning. 2018.
> >
> > [5] Sunehag, Peter, et al. "Value-decomposition networks for cooperative multi-agent learning." arXiv preprint arXiv:1706.05296 (2017).

---

### Official Review · AnonReviewer2 · 2020-10-28
**Downgraded distributional reinforcement learning for multi-agent systems**

**Rating:** 4
**Confidence:** 4

**Review:**

This paper proposes a new value-based method using risk measures in cooperative multi-agent reinforcement learning. The authors propose a new network structure that calculates global CVaR through individual distribution and learns risk-sensitized multi-agent policies. The authors also propose a new dynamic risk level prediction method that can dynamically adjust the risk level according to the agent’s observation and action. Applying risk-sensitive reinforcement learning in multi-agent reinforcement learning is interesting, but several points can be improved.

This paper has a fundamental difference compared to the distributional reinforcement learning recently studied in single-agent reinforcement learning like IQN. In single-agent reinforcement learning, the distribution of Q-function is first learned through the distributional Bellman operator, and the learned distribution can be utilized in additional applications such as risk-sensitive reinforcement learning. Even if the mean value of the distribution is used without considering the risk, it shows higher performance than the existing reinforcement learning algorithms without distribution. The distributional Bellman operator has a richer training signal than the Bellman operator for the existing scalar Q-function, which enables fast representation learning. Moreover, by changing the sampling distribution through the learned distribution, risk-sensitive reinforcement learning can be easily applied to multiple risk measures.

However, in this paper, the authors do not use distribution as a direct learning objective but only aim to maximize CVaR. Unlike the distributional bellman operator, this learning method cannot be expected to improve training speed, and it is difficult to understand the meaning of the learned return distribution learned with the scalar loss function.

Also, this paper proposes a dynamic risk level prediction, but this part is confused. The authors argue that this method solves time-consistency issues and unstable learning. However, there is an insufficient explanation as to why this problem occurred and how to solve it. There is also a lack of explanation on why risk level is divided into K steps and why alpha is defined as in equation (3). Finally, detailed analyses are needed on how dynamic risk level prediction affects performance.

There is a paper, which the authors must cite and compare to. This is not the same as RMIX, but close enough that it has to be compared.

Hu et al. (2020).  QR-MIX: Distributional Value Function Factorisation for Cooperative Multi-Agent Reinforcement Learning. URL:https://arxiv.org/pdf/2009.04197.pdf

---

> ### Author Response · Authors · 2020-11-20
> **Responses to AnonReviewer2 (part 1)**
>
> Dear reviewer, we thank you for your insightful comments on our paper. In the revised version, we cited QR-MIX paper and compared RMIX with QR-MIX on 10 scenarios. The results can be found in Section 4.2 and Appendix G.3. We also present results analysis of RMIX on corridor in Appendix G.2.  We now explain your concerns and answer your questions point by point.
>
> ---
>
> **Q1: the authors do not use distribution as a direct learning objective but only aim to maximize CVaR.**
>
> **A1**: We use CVaR values as policies for decentralized execution in RMIX.
>
> (1) CVaR is a widely used risk measure and enjoys many properties such as convexity. The risk-sensitive RL [1,2,3,4] is an important research problem. In RMIX, each agent aims to estimate a local return distribution at each time step and uses the local return distribution to estimate the CVaR analytically for decentralized execution. RMIX estimates the global CVaR value via the monotonic network.
>
> (2) Risk-sensitive RL and distributional RL are orthogonal research topics. IQN (a distributional RL method) [5] made a connection between distributional RL and risk-sensitive RL by utilizing a risk measure over the return distribution. Distributional RL updates the return distribution while risk-sensitive RL optimizes the risk values (policies).
>
> ---
>
> **Q2: Unlike the distributional bellman operator, this learning method cannot be expected to improve training speed**
>
> **A2**: We answer your question in the following.
>
> (1) RMIX aims to learn risk-sensitive policies for enhanced cooperation and improving the training speed is not our main focus.
>
> (2) RMIX shows slower training speed compared with baselines due to the network architecture. We present training time cost of each algorithm on some scenarios as shown in the table below. Despite this fact, RMIX gains superior performance on corridor (very hard), 6h_vs_8z (very hard),  27m_vs_30m (very hard) and many other scenarios. We train our model on NVIDIA Tesla V100 GPU 16G and Intel(R) Xeon(R) CPU E5-2683 v3 @ 2.00GHz.
>
> | Scenarios      | Training steps | RMIX            | QMIX     | VDN      | IQL      | QTRAN    | MAVEN     |
> | -------------- | -------------- | --------------- | -------- | -------- | -------- | -------- | --------- |
> | corridor       | 3 million      | 1 day 13 hours | 24 hours | 21 hours | 22 hours | 21 hours | 23 hours  |
> | 3s5z\_vs\_3s6z | 3 million      | 23 hours        | 20 hours | 18 hours | 19 hours | 19 hours | 17 hours  |
> | MMM2           | 3 million      | 22 hours        | 20 hours | 18 hours | 19 hours | 19 hours | 16 hours  |
> | 6h\_vs\_8z     | 3 million      | 1 day 18 hours | 20 hours | 19 hours | 19 hours | 20 hours | 15 hours  |
> | 5m\_vs\_6m     | 2 million      | 19 hours        | 18 hours | 9 hours  | 12 hours | 13 hours | 11 hours  |
> | 8m\_vs\_9m     | 1 million      | 8 hours         | 8 hours  | 8 hours  | 8 hours  | 8 hours  | 6 hours   |
> | 10m\_vs\_11m   | 1 million      | 9.5 hours       | 8 hours  | 6 hours  | 7 hours  | 8 hours  | 7.5 hours |
> | 27m\_vs\_30m   | 1.5 million    | 23.5 hours      | 18 hours | 17hours  | 16 hours | 12 hours | 20 hours  |
>
> However, the training speed of Deep RL and MARL algorithms varies on different computational platforms, hyper-parameters and training schemes. For training StarCraft II scenarios, when using high performance CPUs such as Intel(R) Core(TM) i9-9820X CPU @ 3.30GHz, it shows faster training speed especially on scenarios where the number of agent is large, for example 27m_vs_30m.
>
> (3) In fact, RMIX shows good sample efficiency compared with baseline methods (QMIX, MAVEN, Qatten, etc.) when training super hard scenarios such as corridor, 27m_vs_30, 3s5z_vs_5s6z (Figure 6 and 7 in the paper). Improving sample efficiency is vital for RL and MARL.
>
> (4) To our best knowledge, the paper of C51 [6] shows that “for N = 51, our TensorFlow implementation trains at roughly 75% of DQN’s speed” (in footnote 2, page 6 in the C51 paper). So, with Distributional Bellman Operator, distributional RL methods can be slower than DQN. The main reason is updating the return distribution takes more computational resources as the loss function causes the distribution update to requires complex distribution projection as shown in C51’s paper. Besides that modelling distribution adds more parameters in the neural network, which can slow down the training speed.

---

> > ### Author Response · Authors · 2020-11-20
> > **Responses to AnonReviewer2 (part 2)**
> >
> > **Q3: it is difficult to understand the meaning of the learned return distribution learned with the scalar loss function**
> >
> > **A3**: We appreciate your understanding on distributional RL in single-agent domain and distributional value function factorisation ideas in the QR-MIX paper.
> > In fact, with CVaR risk measure selecting some lower value regions of the local return distribution given a risk level, the weights of chosen regions are updated with the monotonic value network in an end-to-end fashion and the weights of regions which are not chosen are not update. This update method has not been investigated but shows practical merits. During training, as agents have collected the experience from the environment in different states, the weights of different regions in the local return distribution are updated given states visited by agents. Although there is a gap between previous works and the updating methods used in RMIX, it shows good results in RMIX, which motivates future research to create a new form of distributional RL methods.
> >
> > ---
> >
> > **Q4: this method solves time-consistency issues and unstable learning. However, there is an insufficient explanation as to why this problem occurred and how to solve it.**
> >
> > **A4**: The time-consistency issue is an issue in risk-sensitive RL [10,11] and it is usually defined as follows [11]: if a certain outcome is considered less risky in all states of the world at stage t + 1, then it should also be considered less risky at stage t. However, a static measure can lead to “time-inconsistent” behaviour [11]. In multi-agent scenarios, this is also a problem due to the non-stationarity where agents take actions independently. Motivated by this, besides static risk measure, we also consider the dynamic risk measure to alleviate these circumstances by designing another network to model agents’ trajectories via GRU to predict the risk level.
> >
> > ---
> >
> > **Q5: why risk level is divided into K steps and why alpha is defined as in equation (3).**
> >
> > **A5**:
> >
> > (1) We discretize the risk level (0, 1) into K for the purpose of CVaR calculation with the local distribution.
> >
> > (2) We agree it is confusing. In Eq. 3, the left side should probability of $P(\alpha_i)$. As we can use Eq. 3 to calculate the probability of each risk level, we can get a probability vector containing K probability values, and we can get the  $k \in [1, …, K]$ with the maximal probability via argmax and normalize $k$ into $(0, 1]$, thus  $\alpha_i$ is calculated.. We can use this value to mask out parts of local return distribution out of the risk level to get the final CVaR value.
> >
> > ---
> >
> > **Q6: detailed analyses are needed on how dynamic risk level prediction affects performance.**
> >
> > **A6**:
> >
> > We added analysis in the paper as shown in Appendix in the revised version.
> >
> > ---
> >
> > **Q7: There is a paper, which the authors must cite and compare to. This is not the same as RMIX, but close enough that it has to be compared.
> > Hu et al. (2020). QR-MIX: Distributional Value Function Factorisation for Cooperative Multi-Agent Reinforcement Learning. URL:https://arxiv.org/pdf/2009.04197.pdf**
> >
> > **A7**:
> >
> > (1) QR-MIX considers decomposing the estimated global (joint) return distribution into individual Q values. However, our RMIX considers the multi-agent risk-sensitive problem, which has rarely been discussed in previous works under the Dec-POMDP framework. The main focuses of RMIX and QR-RMIX are different.
> >
> > (2) We have cited the QR-MIX paper in our paper and implemented QR-MIX to compare QR-MIX with RMIX, though the code of QR-MIX is not open-sourced and the QR-MIX paper has been accepted by any conferences and journals.

---

> > > ### Author Response · Authors · 2020-11-20
> > > **Responses to AnonReviewer2 (part 3)**
> > >
> > > **References**:
> > >
> > > [1] Howard, Ronald A., and James E. Matheson. "Risk-sensitive Markov decision processes." Management science 18.7 (1972): 356-369.
> > >
> > > [2] Marcus, Steven I., et al. "Risk sensitive Markov decision processes." Systems and control in the twenty-first century. Birkhäuser, Boston, MA, 1997. 263-279.
> > >
> > > [3] Chow, Yinlam, et al. "Risk-sensitive and robust decision-making: a cvar optimization approach." Advances in Neural Information Processing Systems. 2015.
> > >
> > > [4] Garcıa, Javier, and Fernando Fernández. "A comprehensive survey on safe reinforcement learning." Journal of Machine Learning Research 16.1 (2015): 1437-1480.
> > >
> > > [5] Dabney, Will, et al. "Implicit Quantile Networks for Distributional Reinforcement Learning." International Conference on Machine Learning. 2018.
> > >
> > > [6] Bellemare, Marc G., Will Dabney, and Rémi Munos. "A Distributional Perspective on Reinforcement Learning." International Conference on Machine Learning. 2017.
> > >
> > > [7] Littman, Michael L. "Markov games as a framework for multi-agent reinforcement learning." Machine learning proceedings 1994. Morgan Kaufmann, 1994. 157-163.
> > >
> > > [8] Lowe, Ryan, et al. "Multi-agent actor-critic for mixed cooperative-competitive environments." Advances in neural information processing systems 30 (2017): 6379-6390.
> > >
> > > [9] Hu, Jian, et al. "QR-MIX: Distributional Value Function Factorisation for Cooperative Multi-Agent Reinforcement Learning." arXiv preprint arXiv:2009.04197 (2020)..
> > >
> > > [10] Ruszczyński, Andrzej. "Risk-averse dynamic programming for Markov decision processes." Mathematical programming 125.2 (2010): 235-261.
> > >
> > > [11] Tamar, Aviv, et al. "Policy gradient for coherent risk measures." Advances in Neural Information Processing Systems. 2015.

---

> > > > ### Comment · AnonReviewer2 · 2020-11-23
> > > > **Thank you for the kind responses. I still have questions.**
> > > >
> > > > - In Q2, the meaning of "training speed" I intended was "sample efficiency". Sorry for your confusion. But your answers helped me understand.
> > > >
> > > > - Thank you for the answer to Q3. I am still curious about how distribution is learned. This is because the learning method looks quite different from the traditional risk-sensitive RL or distributional RL. What you're talking about is that distribution is learned by this because only a few of the Dirac delta functions are selected, right? This region-specific update method seems to work well for dynamic risk, but it does not seem to work well for static risk levels. Is it correct?
> > > >
> > > > - The experimental results in this paper are very positive, but I'm curious if this really happened because of "risk-averse".
> > > > Because of the static risk levels in the experiment, risk-neutral seems to be the best. I think a more in-depth comparison with LH-IRQN [1] is needed. This paper talks about the need for a "risk-seeking" strategy in cooperative multi-agent problems. I am very pleased with the claims in this paper. In MARL, agents may benefit from risk-seeking policies as seeking the highest possible utility help the team breaks out of sub-optimal shadowed equilibria. In conclusion, It would be nice if there are results when using CVnaR used by LH-IRQN instead of CVaR even in a simple environment like 8m_vs_9m.
> > > >
> > > > - Do you have any plans to release the code?

---

> > > > > ### Author Response · Authors · 2020-11-23
> > > > > **Response to AnonReviewer2's new questions**
> > > > >
> > > > > **Comment 0: In Q2, the meaning of "training speed" I intended was "sample efficiency". Sorry for your confusion. But your answers helped me understand.**
> > > > >
> > > > > **A0:**
> > > > >
> > > > > We are very glad that our response addressed your concerns.
> > > > >
> > > > > ---
> > > > >
> > > > > **Q1: What you're talking about is that distribution is learned by this because only a few of the Dirac delta functions are selected, right? This region-specific update method seems to work well for dynamic risk, but it does not seem to work well for static risk levels. Is it correct?**
> > > > >
> > > > > **A1:**
> > > > >
> > > > > Your understanding of the updating method is correct (we provided an answer A3 of this anonymous link https://openreview.net/forum?id=1EVb8XRBDNr&noteId=-1PhgrZuou). In general, in our experiments, our method with static risk levels also superior results over QMIX as shown in Figure 10.
> > > > >
> > > > > ---
> > > > >
> > > > > **Q2: The experimental results in this paper are very positive, but I'm curious if this really happened because of "risk-averse". Because of the static risk levels in the experiment, risk-neutral seems to be the best.**
> > > > >
> > > > > **A2:**
> > > > >
> > > > > RMIX agent selects an action via CVaR values with dynamic risk levels. The risk-neutral policies are based on expectation over the return distribution. In our paper, when the $\alpha=1$, it is risk-neutral, RMIX ($\alpha=1$). As shown in Figure 10 and 22, by capturing various outcomes, the RMIX ($\alpha=1$) shows good results over QMIX due to capturing richer information, however, it is not as good as RMIX in many scenarios in general.
> > > > >
> > > > > ---
> > > > >
> > > > > **Q3: I think a more in-depth comparison with LH-IRQN [1] is needed. This paper talks about the need for a "risk-seeking" strategy in cooperative multi-agent problems. I am very pleased with the claims in this paper. In MARL, agents may benefit from risk-seeking policies as seeking the highest possible utility help the team breaks out of sub-optimal shadowed equilibria. In conclusion, It would be nice if there are results when using CVnaR used by LH-IRQN instead of CVaR even in a simple environment like 8m_vs_9m.**
> > > > >
> > > > > **A3:**
> > > > >
> > > > > If we are not wrong, the LH-IRQN is from paper [1]. We would like to compare LH-IRQN with our methods. However, experimental details are not clear enough in [1], and the code of LH-IRQN is not available. LH-IRQN cannot be quickly implemented and further trained within one day.
> > > > >
> > > > > LH-IRQN uses risk-seeking policies. In 8m_vs_9m, the number of MARL agent is 8 and correspondingly the number of enemies is 9. This is an asymmetric scenario. When one MARL agent dies, the asymmetricity would be a severe issue for MARL training. As a consequence, it may not proper to train risk-seeking policies. The primal goal of each MARL agent is to keep alive and avoid being killed by the enemies. For grid scenarios in [1], there is no such setting, so risk-seeking policies can help to explore and improve the performance.
> > > > >
> > > > > ---
> > > > >
> > > > > **Q4: Do you have any plans to release the code?**
> > > > >
> > > > > **A4:**
> > > > >
> > > > > For reproducibility, we will release our code after the acceptance of our paper.
> > > > >
> > > > > ---
> > > > >
> > > > > **References:**
> > > > >
> > > > > [1] Lyu, Xueguang, and Christopher Amato. "Likelihood Quantile Networks for Coordinating Multi-Agent Reinforcement Learning." Proceedings of the 19th International Conference on Autonomous Agents and MultiAgent Systems. 2020.

---

> > > > > > ### Comment · AnonReviewer2 · 2020-11-23
> > > > > > **Additional comments**
> > > > > >
> > > > > > Sorry to keep asking. But I want a complete understanding and to make sure that this is not just due to increased neural network capacity.
> > > > > >
> > > > > > - For Q1, I want a more fundamental understanding instead of experiments. However, if RMIX's alpha is static at 1, all weights are always chosen and CVaR will be simply expectation. How can I learn the "distribution" through this? For example, Implicit Quantile Networks learns the distribution with quantile regression and sampling even if they use a risk-neutral policy.  In my opinion, CVaR value of RMIX with static alpha=1 (risk-neutral) is not an expectation of distribution, but rather a Q-value calculation through more features and network parameters.
> > > > > >
> > > > > > - For Q2, what I was talking about was a comparison with other static alphas. Sorry for your confusion.
> > > > > >
> > > > > > - For Q3, I understand that part too. I mean using CVnaR instead of CVaR in RMIX. Wouldn't this be possible by simply changing the direction of the inequality of CVaR? I'm just curious about the RMIX results of the risk-seeking version.  I also understand if you don't have enough time for this experiment.

---

> > > > > > > ### Author Response · Authors · 2020-11-24
> > > > > > > **Response to AnonReviewer2's additional comments**
> > > > > > >
> > > > > > > Dear AnonReviewer2,
> > > > > > >
> > > > > > > More discussions and suggestions on further improving the paper are always welcomed! We hope our response can address your concerns.
> > > > > > >
> > > > > > > ---
> > > > > > >
> > > > > > > **Q1: For Q1, I want a more fundamental understanding instead of experiments. However, if RMIX's alpha is static at 1, all weights are always chosen and CVaR will be simply expectation. How can I learn the "distribution" through this? For example, Implicit Quantile Networks learns the distribution with quantile regression and sampling even if they use a risk-neutral policy. In my opinion, CVaR value of RMIX with static alpha=1 (risk-neutral) is not an expectation of distribution, but rather a Q-value calculation through more features and network parameters.**
> > > > > > >
> > > > > > > **A1:**
> > > > > > >
> > > > > > > When $\alpha=1$, the distributions are also learned by feeding all the local return distributions into the monotonic network to update weights of agents (note that parameters of agent network are shared by all agents). The monotonic network is designed for assigning and updating weights of distribution values for credit assignment as each agent’s exact contribution to the global reward is unknown. Therefore, the distribution is learned and it captures more information even with the mean values.
> > > > > > >
> > > > > > > We also provide an anonymous video (https://youtu.be/nW7xYui47bI, also available on https://sites.google.com/view/rmix) to show the distributions are learned.
> > > > > > >
> > > > > > > ---
> > > > > > >
> > > > > > > **Q2:For Q2, what I was talking about was a comparison with other static alphas. Sorry for your confusion.**
> > > > > > >
> > > > > > > **A2:**
> > > > > > >
> > > > > > > You can find these experimental results on Figure 16, page 22 in the paper. These experimental results were trained on 6h_vs_8z (super hard), corridor (super hard), 3s5z_vs_3s6z (super hard) and 5m_vs_6m (hard). You can also refer to Table 1 (in page 19) for more detailed introduction of SMAC scenarios.
> > > > > > >
> > > > > > > ---
> > > > > > >
> > > > > > > **Q3: For Q3, I understand that part too. I mean using CVnaR instead of CVaR in RMIX. Wouldn't this be possible by simply changing the direction of the inequality of CVaR? I'm just curious about the RMIX results of the risk-seeking version. I also understand if you don't have enough time for this experiment.**
> > > > > > >
> > > > > > > **A3:**
> > > > > > >
> > > > > > > We would like to try it and put our results if it is possible.

---

> > > > > > > > ### Comment · AnonReviewer2 · 2020-11-24
> > > > > > > > **Additional comments**
> > > > > > > >
> > > > > > > > **Q1**
> > > > > > > >
> > > > > > > > Thanks for your hard work, but sorry I haven't understood it yet. The monotonic network makes learning the expectation of individual values equal to the expected Q-value. The paper claims it is a ***probability distribution***. However, I feel that additional techniques are needed to make the individual values a ***distribution***. Even the video you uploaded cannot prove that it is a ***distribution***.
> > > > > > > >
> > > > > > > > **Q2**
> > > > > > > >
> > > > > > > > I looked at figure 16 when writing the previous reviews and asked why alpha=1 is the best among static alphas. Then I asked, "risk-neutral seems to be the best." But there was a comparison with dynamic alphas, not other static alphas, as an answer. So I said, "I was talking about was a comparison with other static alphas" for the next response.  Could I ask again why 1.0 is the best static alpha? The paper does not compare results between static alphas. I wonder if the risk-averse strategies really help. Because unlike the claim in the first paragraph on page 9 (In the asymmetric scenario, the performance decreases when it is risk-neutral), the performance increases as the alpha increases, as shown in Figure 16.
> > > > > > > >
> > > > > > > > Thanks again for your responses.

---

> > > > > > > > > ### Author Response · Authors · 2020-11-24
> > > > > > > > > **Response to Additional Comments by AnonReviewer2 (part 1)**
> > > > > > > > >
> > > > > > > > >
> > > > > > > > > Dear reviewer,
> > > > > > > > >
> > > > > > > > > We thank your in-depth thinking on our paper. We now explain your concerns and answer your questions point by point.
> > > > > > > > >
> > > > > > > > > We also present the result of RMIX with CVnAR on https://sites.google.com/view/rmix
> > > > > > > > >
> > > > > > > > > ---
> > > > > > > > >
> > > > > > > > > **Q1: Thanks for your hard work, but sorry I haven't understood it yet.  The monotonic network makes learning the expectation of individual values equal to the expected Q-value. The paper claims it is a probability distribution. However, I feel that additional techniques are needed to make the individual values a distribution. Even the video you uploaded cannot prove that it is a distribution.**
> > > > > > > > >
> > > > > > > > > **A1:**
> > > > > > > > >
> > > > > > > > > Thanks for your feedback.
> > > > > > > > > * Let’s recall the training procedure of RMIX:
> > > > > > > > >
> > > > > > > > >     a). We use networks to output the distribution of Q values for agents
> > > > > > > > >
> > > > > > > > >     b). Then, we compute the CVaR values of each agents with the distribution of Q values with static/dynamic risk levels.
> > > > > > > > >
> > > > > > > > >     c). Then, the CVaR values of agents (rather than the distribution) are fed into the monotonic mixing network to obtain the total value
> > > > > > > > >
> > > > > > > > >     d). For the training, we calculate the loss based on the total value and backpropagate it to agents’ networks as well as the risk level module in dynamic risk-level variant. This is the exactly training procedure of the most MARL methods, e.g., QMIX, QTRAN.
> > > > > > > > >
> > > > > > > > > * Therefore, we would like to note the following aspects:
> > > > > > > > >
> > > > > > > > >     a). The fundamental motivation of RMIX is to use risk-sensitive measure, i.e., CVaR, into MARL. The distributions learned by distributional RL only facilitate the computation of CVaR values in single-agent domain. Therefore, it is the CVaR values rather than the distributions of Q values are fed into the mix network.
> > > > > > > > >
> > > > > > > > >     b). To make agents perform well, RMIX trains both the networks for agents (which output the local distribution of Q values) and the mix network, as well as the risk level module in the dynamic risk-level variant of RMIX.
> > > > > > > > >
> > > > > > > > > * Our uploaded video demonstrates that the networks for agents are trained, i.e., the distributions of Q values for agents are trained as Dirac functions are initialized by sampling values and then decomposed global rewards are used to update each Dirac functions. The empirical results in our paper demonstrate that our methods outperform existing methods, which indicate that leveraging risk-sensitive measures in MARL is important.
> > > > > > > > >
> > > > > > > > > * We will consider feeding the distributions of Q values of agents into the mix network in the future, which is beyond the scope of this paper, i.e., using risk-sensitive measure into MARL. RMIX is one implementation of risk-sensitive MARL under the Dec-POMDP framework. As there are few works have been done, there is much work to be done both in theoretically and practically. We take the first step in this problem and we hope more attention should be paid on this problem to further facilitate the application of MARL in real-world problems.

---

> > > > > > > > > ### Author Response · Authors · 2020-11-24
> > > > > > > > > **Response to Additional Comments by AnonReviewer2 (part 2)**
> > > > > > > > >
> > > > > > > > > **Q2: I looked at figure 16 when writing the previous reviews and asked why alpha=1 is the best among static alphas. Then I asked, "risk-neutral seems to be the best." But there was a comparison with dynamic alphas, not other static alphas, as an answer. So I said, "I was talking about was a comparison with other static alphas" for the next response.**
> > > > > > > > >
> > > > > > > > > **A2:**
> > > > > > > > >
> > > > > > > > > We did not know the results you mentioned were in Figure 16 as there are many figures in our paper and you did not mention the figure 16 in your previous questions.
> > > > > > > > >
> > > > > > > > > ---
> > > > > > > > >
> > > > > > > > > **Q3: Could I ask again why 1.0 is the best static alpha? The paper does not compare results between static alphas. I wonder if the risk-averse strategies really help. Because unlike the claim in the first paragraph on page 9 (In the asymmetric scenario, the performance decreases when it is risk-neutral), the performance increases as the alpha increases, as shown in Figure 16.**
> > > > > > > > >
> > > > > > > > > **Q3.1: Could I ask again why 1.0 is the best static alpha?**
> > > > > > > > >
> > > > > > > > > **A:**
> > > > > > > > >
> > > > > > > > > (i): $\alpha=1$ is not the best compared with other static $\alpha$ values in some cases.
> > > > > > > > >
> > > > > > > > > We also present results on Figure 16 with static values to show learning time-consistency $\alpha$ values can help to improve the performance of static $\alpha \in (0, 1)$ as we claim in the paper.
> > > > > > > > >
> > > > > > > > > With increasing $\alpha$ values to 1, it is much stable to train policies for many scenarios and training these hard scenarios takes many days and many computational resources. So, for $\alpha \in (0.7, 1]$, we did not show them in the paper. We will put them in the final version if they help to enhance the understanding of the paper.
> > > > > > > > >
> > > > > > > > > (ii): For static $\alpha$ values ($\alpha < 1$), it depends on environments as in some states, $\alpha$ values should not be static for risk-averse policies. In general, with $\alpha=1$, it is easier for training. We do not claim that 1.0 is the best static $\alpha$. We present several results to show that the values of alpha influence the agents’ performance. But we cannot do a sweeping to for all static alpha values. Therefore, we introduce the dynamic risk level to dynamically determine the risk levels to improve the performance of agents and alleviate the time-consistency issue.
> > > > > > > > >
> > > > > > > > > **Q3.2: “I wonder if the risk-averse strategies really help.”**
> > > > > > > > >
> > > > > > > > > **A:**
> > > > > > > > >
> > > > > > > > > Risk-averse strategies help in many scenarios, especially for hard asymmetric scenarios, for example, 3s5z_vs_3s6z and 5m_vs_6m. However, as we discussed in the paper and in the previous comments, there is a time-consistency problem and static risk level may impede the convergence and we introduced the dynamic $\alpha$ values.
> > > > > > > > >
> > > > > > > > > We do the ablation experiments to show that by dynamically changing the risk level, the agents’ performance can be increased, which indicate that the different risk-averse strategies can really help agents.

---

### Author Response · Authors · 2020-11-20
**General Response**

We first thank all the reviewers for their constructive and valuable comments. We really appreciate positive comments made by reviewers who recognised our contribution to MARL. We briefly summarize our updates:

1. Updated the paper (updates are in red), including:

    (a). Cite QR-MIX paper in Section 1;

    (b). Implement code of QR-MIX (https://arxiv.org/pdf/2009.04197.pdf) and present results of RMIX vs QR-MIX on 10 StarCraft II scenarios (6 scenarios compared in QR-MIX paper). Experiment results are shown in Figure 9 and 18, page 25;

    (c). Clarify the risk level predictor in Section 3.2 and move one figure (now Figure 4) from Appendix to Section 3.2;

    (d). Add pseudo-code of RMIX in Appendix D;

    (e). Present RMIX results analysis on Corridor in Appendix G.2, page 24;

    (f). Present results and game replay of RMIX on an anonymous link: https://sites.google.com/view/rmix;

2. Updated response as per the reviews;

We hope the responses could address the comments of all the reviewers. We sincerely hope that the reviewers can re-evaluate our paper after reading our updated version. More discussions and suggestions on further improving the paper are also always welcomed!

---

> ### Comment · Area_Chair1 · 2020-11-21
> **Reviewers, please read responses and update**
>
> Dear reviewers,
>
> Thank you very much for your reviews.  The authors have now provided detailed responses to your concerns & questions and revised the paper.  Please read those to see whether your concerns are resolved or you still see remaining issues.
>
> Dear authors,
>
> Thank you very much for your responses, which I have quickly read together with the paper and reviews.  I would like to ask a few questions, which might help reviewers as well.

---

> > ### Comment · Area_Chair1 · 2020-11-21
> > **Q1 (Theorem 1)**
> >
> > Reviewer 1 and 4 had questions on Theorem 1.  It would help to clarify why Theorem 1 does not hold for the following case, and what assumptions are needed to exclude this case from Theorem 1.
> >
> > Consider two agents, 1 and 2.  Each agent has two possible actions, "risky" and "safe".  With risky action, reward is 0 with probability 0.5, and 10 otherwise. With safe action, reward is 1 with probability 1.  Let X(i,a) be the random variable representing the reward with action a for agent i.  Total reward is Y(a1,a2) = X(1,a1) + X(2,a2), and we assume X(1,a1) and X(2,a2) are independent.
> >
> > Assume that the risk-sensitivity is alpha=0.5.  Then CVaR0.5[X(i,risky)] = 0 and CVaR0.5[X(i,safe)] = 1, so each agent would chose "safe" to individually maximize CVaR0.5.  However, CVaR0.5[Y(risky,risky)] = 5, CVaR0.5[Y(risky,safe)] = CVaR0.5[Y(safe,risky)] = 1, and CVaR0.5[Y(safe,safe)] = 2.  So, both taking "risky" would maximize the CVaR0.5 of total reward.

---

> > > ### Author Response · Authors · 2020-11-23
> > > **Response to Q1 (Theorem 1)**
> > >
> > > **A1:**
> > >
> > > We agree that the case presented in the question is right. The case in the question conducts CVaR over the over the joint reward distribution of individuals’ reward distributions. The difference is that, Theorem 1 takes the individual’s CVaR values as input by $C^{\operatorname{tot}} (\boldsymbol{\tau}, \boldsymbol{u}, \boldsymbol{\alpha}) = f_{\operatorname{m}}(C_{1}(\tau_{1}, u_{1}, \alpha_{1}), \dots, C_{n}(\tau_{n}, u_{n}, \alpha_{n}))$ with monotonicity network $f_m$.
> > >
> > > Therefore, we do not use CVaR over the global return distribution and instead resort to $C^{\operatorname{tot}}$ the surrogate global CVaR as our optimization target to optimize each individual CVaR values. We provide reasons:
> > >
> > > (i) Rewards in StarCraft II are deterministic. In the case provided in the question, as reward distribution of the “safe” action is deterministic, it is subtle that the CVaR is not defined in practice and the stochasticity is certain;
> > >
> > > (ii) Even though the model underlying the environment provides deterministic rewards, such deterministic reward comes from various cooperation strategies as the policies of the agent are not certain (stochastic policies, exploitation vs exploration, etc.) and contributions of each agent to the global reward are unknown.

---

> > ### Comment · Area_Chair1 · 2020-11-21
> > **Q2 (Dynamic risk level)**
> >
> > All of the reviewers had shown some concerns or confusions about the dynamic risk level.  I would like to ask a question related to the one asked by Reviewer 1.  I understand that the value of alpha is dynamically adjusted, but what criteria is used to determine which values are good?

---

> > > ### Author Response · Authors · 2020-11-23
> > > **Response to Q2 (Dynamic risk level)**
> > >
> > > **A2:**
> > >
> > > The best criteria of $\alpha_{i,t}$ at time step $t$ for agent $i$ is that $\alpha_{i,t}$ should stick to $\alpha_{i,t-1}$ if the agent is still in the same decision stage of last time step. Decision stages are consecutive observations separated by time during the episode. Observations in the same decision stage are similar which makes the risk level similar and thus the $\alpha$ values are similar. The decision stages are not predefined but learned by the two GRUs.
> > >
> > > Concretely, in order to calculate the alpha, with $\tau_{0:t}$ and $u^{t-1}$ as input we use one GRU together with the agent network to generate current local distribution for each agent. With $\tau_{0:t-1}$ and $u^{t-2}$ as input, the other GRU and the network in the risk operator is used to generate the local distribution of the last step as shown in Figure 2(a). We use inner product of the two local distributions to measure the discrepancy between them and the discrepancy vector is a $K$-dimension vector and softmax is used to select the level of $\alpha$ via argmax and then normalize it in to (0, 1] as shown in Figure 4.
> > >
> > > We also present an example in Figure 17, page 24 in the revised paper (you can also find it on the anonymous site of our paper: https://sites.google.com/view/rmix) to show this setting.

---

> > ### Comment · Area_Chair1 · 2020-11-21
> > **Q3 (Time-consistency)**
> >
> > Reviewer 2 had some confusion about time-consistency.  I have a related question.  The Bellman-like equation shown in (6) does not hold when C^tot=CVaR, because CVaR is not time-consistent.  What role does (6) play in the proposed approach?

---

> > > ### Author Response · Authors · 2020-11-23
> > > **Response to Q3 (Time-consistency)**
> > >
> > > **A3:**
> > >
> > > **(i) Question on Eqn. (6):**
> > >
> > > **A(i)**
> > >
> > > Eqn. (6) is for defining the **temporal-difference (TD) objective with the surrogate global CVaR** for training method and during training, all information, including individuals’ observations, global rewards and global states and trajectories of all agents, can be used to train the Q values/policies of each agent for decentralized execution.
> > >
> > > The motivation behind Eqn. (6) is that our goal is to design a risk-sensitive MARL approach under the Dec-POMDP framework and we use CVaR values as the policies of the agents instead of Q values. During training, individuals’ CVaR values are used as input into (central) monotonic network for centralized training.
> > >
> > > **(ii) Question on time-consistency:**
> > >
> > > **A(ii)**
> > >
> > > You are right, as you commented, unlike the expected value, the CVaR objective is not time-consistent, which can lead to non-stationary policies in single-agent RL and MARL. In our paper, to alleviate it, we use two separate GRUs (one for the return distribution estimator and another for the risk level predictor as shown in Figure 3 in our paper) to model trajectories of an episode and use the hidden state with the current observation and last step’s action as input in order the expand the state space to capture the historical trajectory.

---

> > ### Author Response · Authors · 2020-11-23
> > **General Response to Area Chair1 and Reviewers**
> >
> > Dear Area Chair1,
> >
> > Thank you for handling our submission and providing valuable and perceptive comments on our paper. We appreciate the example presented in the question which is very good for us to gain a deeper understanding of our method. We hope our responses could explain your questions. More discussions and suggestions on further improving the paper are also always welcomed!
> >
> >
> > Dear reviewers,
> >
> > We hope our responses to questions from Area Chair1 could also address the comments and clarify our ideas.

---

### Author Response · Authors · 2020-11-25
**Summary of updates before the end of author discussion period**

Dear Reviewers and Area Chair,

We first thank all valuable, constructive and perspective comments made by the Area Chair and all reviewers to facilitate the two-week rebuttal, and further help improving our paper. We are glad that our previous responses have addressed some concerns.

As the rebuttal system will be closed for authors in a few hours, we briefly summarize our new updates after discussing with the AnonReviewer2:

* We provided an anonymous video (https://youtu.be/nW7xYui47bI, also available on https://sites.google.com/view/rmix) to show the distributions are learned and they capture more information even with the mean values. The data used is from RMIX trained on 3m.
* We also presented the results of RMIX vs RMIX with Conditional Value no at Risk (RMIX-CVnaR, risk-seeking policies) on 2m_vs_1z (available on https://sites.google.com/view/rmix). The results show that RMIX-CVnaR is not as good as RMIX even in this simple scenario.

Thanks again.

Sincerely yours,

Authors of Paper207

---

### Decision · Program_Chairs · 2021-01-07
**Final Decision**

**Decision:**

Reject

**Comment:**

This paper proposes a method of risk-sensitive multi-agent reinforcement learning in cooperative settings.  The proposed method introduces several new ideas, but they are not theoretically well founded, which has caused many confusions among the reviewers.  Although some of the confusions are resolved through discussion, there remain major concerns about the validity of the method.